# A mesocortical glutamatergic pathway modulates neuropathic pain independent of dopamine co-release

Miao Li [1] & Guang Yang [1] ✉

Dysfunction in the mesocortical pathway, connecting the ventral tegmental area (VTA) to the prefrontal cortex, has been implicated in chronic pain. While extensive research has focused on the role of dopamine, the contribution of glutamatergic signaling in pain modulation remains unknown. Using in vivo calcium imaging, we observe diminished VTA glutamatergic activity targeting the prelimbic cortex (PL) in a mouse model of neuropathic pain. Optogenetic activation of VTA glutamatergic terminals in the PL alleviates neuropathic pain, whereas inhibiting these terminals in naïve mice induces pain-like responses. Importantly, this pain-modulating effect is independent of dopamine co-release, as demonstrated by CRISPR/Cas9-mediated gene deletion. Furthermore, we show that VTA neurons primarily project to excitatory neurons in the PL, and their activation restores PL outputs to the anterior cingulate cortex, a key region involved in pain processing. These findings reveal a distinct mesocortical glutamatergic pathway that critically modulates neuropathic pain independent of dopamine signaling.

The ventral tegmental area (VTA) is a key brain region involved in reward and motivated behaviors[1–3]. As part of the mesocortical pathway, VTA dopaminergic neurons project to the prefrontal cortex (PFC)[2,4], where dopamine signaling plays a pivotal role in reward-based learning[5,6]. This dopaminergic pathway from the VTA to the PFC has also been implicated in processing aversive experiences, including nociception[2,7]. Studies in both humans and animals have reported reduced neuronal activity in the VTA under chronic pain conditions[8,9]. Furthermore, enhancing dopamine transmission from the VTA to the PFC has been shown to alleviate behavioral deficits associated with neuropathic pain in mice[10].

While previous research has predominantly focused on the role of dopaminergic neurons in pain regulation, it is important to note that the VTA comprises a substantial population of glutamatergic neurons that also project to the PFC[4]. Earlier studies have demonstrated that VTA glutamatergic neurons project to the nucleus accumbens (NAc) and lateral habenular, contributing to reward, aversion, and chronic pain[11–13]. However, the specific role of the VTA–PFC glutamatergic pathway in pain modulation remains unclear. Adding to the complexity, it has been reported that some VTA glutamatergic neurons can

co-release dopamine[14,15], although this phenomenon appears to be specific to certain projections[16]. For instance, within the mesocortical pathway, approximately two-thirds of VTA neurons projecting to the PFC express the type-2 vesicular glutamate transporter (vGluT2), and about 40% of these neurons co-express tyrosine hydroxylase (TH), an enzyme involved in dopamine biosynthesis[15]. Interestingly, despite the presence of both glutamate and dopamine, VTA glutamatergic neurons have been shown to drive reinforcement behavior independently of dopamine co-release[17]. These findings suggest that VTA–PFC glutamatergic transmission may modulate pain signals separately from dopamine activity.

In this study, we investigated the independent role of the VTA–PFC glutamatergic pathway in pain modulation using a mouse model of neuropathic pain. Previous studies in this model have shown that alterations in the prelimbic cortex (PL), a subdivision of the PFC, and its downstream partner, the anterior cingulate cortex (ACC), contribute to enhanced nocifensive behavior[18–22]. Using in vivo two-photon $Ca^{2+}$ imaging, we observed a significant decrease in VTA glutamatergic activity within the PL under neuropathic pain conditions. Optogenetic activation of VTA glutamatergic terminals in the PL

[1]Department of Anesthesiology, Columbia University Irving Medical Center, New York, NY 10032, USA. ✉e-mail: gy2268@cumc.columbia.edu

alleviated neuropathic pain-associated behaviors, while inhibition of these projections in naïve mice elicited pain-like behaviors. To specifically investigate the role of dopamine co-release, we employed a viral-based CRISPR/Cas9 approach to disrupt dopamine synthesis in VTA neurons. Our results demonstrated that VTA–PL glutamatergic projections modulate pain perception independently of dopamine co-release. Additionally, we provided evidence that VTA inputs preferentially target excitatory neurons in the PL. Activation of VTA–PL projections restored PL outputs to ACC and attenuated neuropathic pain-associated behaviors. Together, our findings demonstrate the involvement of a distinct mesocortical glutamatergic pathway in pain modulation.

## Results

### VTA–PL glutamatergic activity is reduced in mice with neuropathic pain

To identify VTA glutamatergic neurons projecting to the PL, we injected an FLP-dependent retrograde transducing adeno-associated virus (AAVrg) encoding Cre into the PL of *Vglut2*[IRES-FLPo] mice, along with a Cre-dependent AAV expressing mCherry into the VTA (Fig. 1a). This injection strategy allowed for the selective expression of mCherry in VTA vGluT2[+] cells projecting to the PL (Fig. 1b). Consistent with previous reports[15], we found that these VTA–PL vGluT2[+] neurons were predominantly located in the anterior part of VTA (-82.05 ± 2.16%) (AP

−2.92 ~ −3.40 mm) (Fig. 1c). Subsequently, we specifically infected VTA glutamatergic neurons with a Cre-dependent AAV encoding cytoplasmic tdTomato and synaptophysin-fused EGFP (Fig. 1d). Confocal imaging in the PL revealed tdTomato-expressing axons with numerous EGFP puncta (Fig. 1e), indicating the presence of presynaptic terminals of VTA glutamatergic neurons in the PL.

To investigate alterations in VTA–PL glutamatergic inputs associated with neuropathic pain, we used in vivo two-photon calcium imaging to examine the axonal activity of VTA glutamatergic neurons in the PL of awake, head-restrained mice[23,24]. For this purpose, an AAV encoding axon-targeted GCaMP6s under Cre-dependent control was injected into the VTA of *Vglut2*[IRES-Cre] mice (Fig. 1f). Subsequently, the mice underwent spared nerve injury (SNI) to induce persistent neuropathic pain[25,26]. Two weeks after surgery, corresponding to the chronic phase of neuropathic pain, we observed a ~50% reduction in Ca[2+] activity within the axons of VTA glutamatergic neurons in resting SNI mice compared to sham mice (14.43 ± 0.56% *vs.* 25.40 ± 0.80%, $P < 0.0001$, Fig. 1g, h). Moreover, a larger fraction of VTA vGluT2[+] axons in the PL exhibited lower levels of activity ($\Delta F/F_0 < 15\%$) in SNI mice relative to sham mice (64% *vs.* 18%) (Fig. 1i), suggesting diminished glutamatergic inputs from the VTA to the PL under chronic pain conditions. This reduction in VTA–PL glutamatergic axonal activity was evident in both male and female mice with neuropathic pain (Supplementary Fig. 1a, b). Consistent with spontaneous activity,

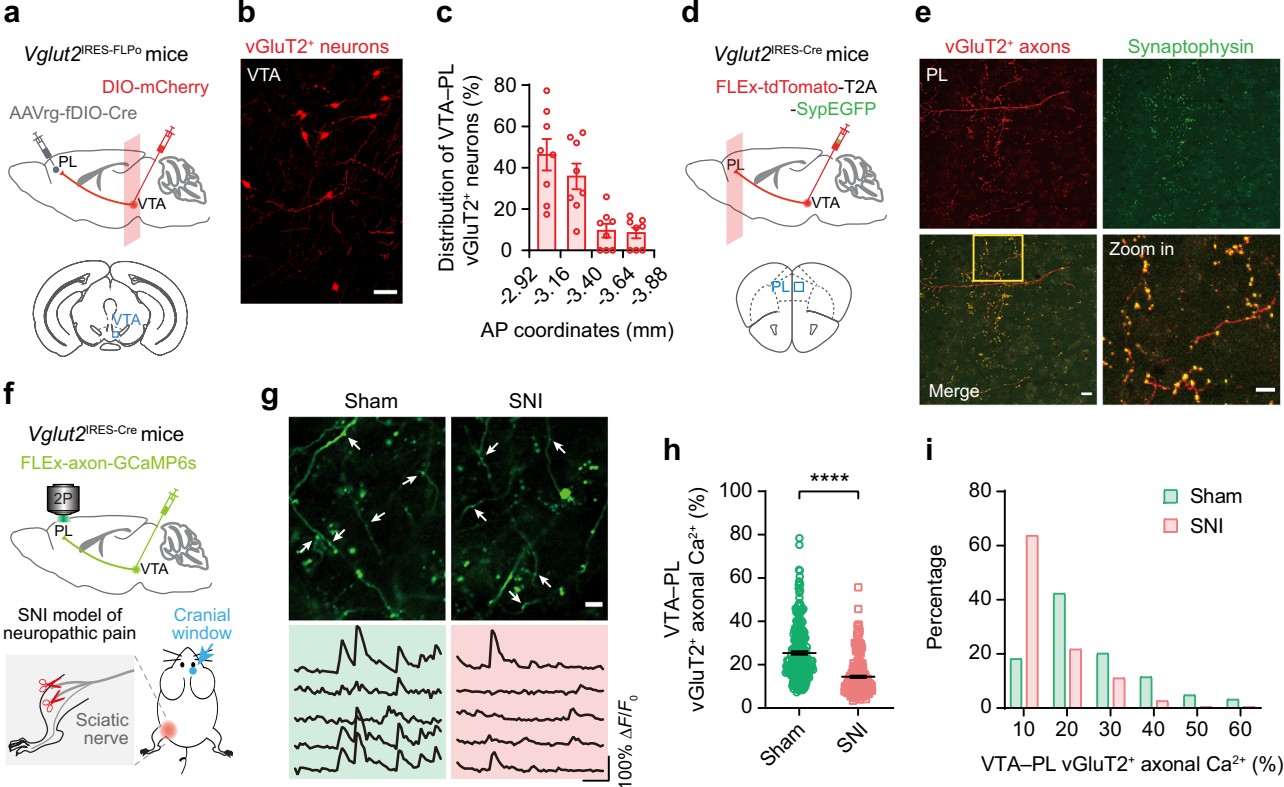

**Fig. 1 | VTA–PL glutamatergic activity is reduced in mice with neuropathic pain.** **a** Experimental design to selectively express mCherry in VTA glutamatergic neurons projecting to the PL. Coronal sections containing VTA are used for confocal imaging. **b** Representative confocal image showing mCherry-expressing VTA glutamatergic neurons projecting to PL. Scale bar, 50 μm. **c** Distribution of mCherry-labelled somas along the rostro-caudal axis of VTA. *n* = 8 mice. **d** Experimental design to virally express tdTomato and presynaptic (synaptophysin-fused) EGFP in VTA glutamatergic neurons. Coronal sections containing PL are used for confocal imaging. **e** Confocal imaging in the PL showing EGFP-labelled presynaptic puncta and tdTomato-labeled axons from VTA glutamatergic neurons (4 mice). Scale bar,

20 μm and 10 μm (zoom in). **f** Experimental design for expressing axon-targeted GCaMP6s in VTA–PL glutamatergic neurons and in vivo two-photon (2P) Ca[2+] imaging in the PL contralateral to the spared nerve injury (SNI) side.
**g** Representative images and fluorescence traces of GCaMP6-expressing axons derived from VTA glutamatergic neurons. Scale bar, 10 μm. **h** Ca[2+] activity in VTA–PL glutamatergic terminals two weeks after sham or SNI surgery ($P < 0.0001$; *n* = 255, 232 axon segments from six mice per group). **i** Distribution plot of data shown in (**h**). Summary data are presented as mean ± S.E.M. ****$P < 0.0001$; by two-sided Mann–Whitney test. Source data are provided as a Source Data file.

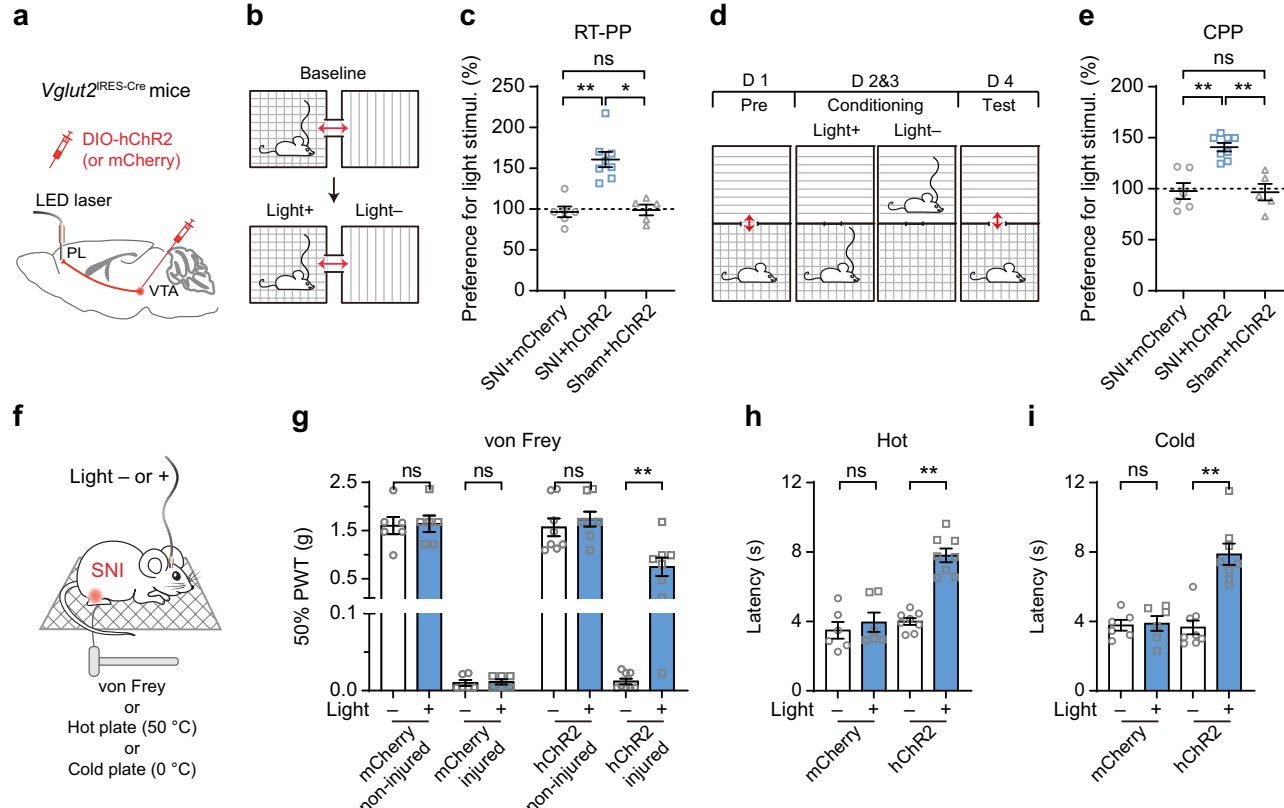

**Fig. 2 | Optogenetic activation of VTA glutamatergic terminals in the PL alleviates pain-associated behaviors. a** Experimental design to express hChR2 or mCherry (control) in VTA glutamatergic neurons and stimulate their terminals in the PL. **b** Schematic of real-time place preference (RT-PP) test for assessing ongoing pain. **c** Preference to stay in the light stimulation-paired chamber for SNI mice expressing mCherry or hChR2 and sham mice expressing hChR2 ($P = 0.0036$ and $0.0137$ for SNI+hChR2 *vs.* SNI+mCherry and sham+hChR2, respectively). **d** Schematic of conditioned place preference (CPP) test for assessing ongoing pain. **e** Preference to stay in the light stimulation-paired chamber for SNI mice expressing mCherry or hChR2 and sham mice expressing hChR2 ($P = 0.0077$ and $0.0061$ for

SNI+hChR2 *vs.* SNI+mCherry and sham+hChR2, respectively). In (**c**, **e**), $n = 6, 8, 5$ mice. **f** Schematic of mechanical and thermal tests with/without light stimulation in SNI mice. **g**–**i** Measurements of nociceptive thresholds in SNI mice expressing mCherry or hChR2. Activation of VTA−PL glutamatergic terminals increases the animals' mechanical ($P = 0.0078$), hot ($P = 0.0078$), and cold ($P = 0.0078$) thresholds in the limb ipsilateral to SNI. In **g**–**i** $n = 6, 8$ mice for mCherry, hChR2 respectively. Summary data are presented as mean ± S.E.M. *$P < 0.05$, **$P < 0.01$; ns not significant; by Kruskal-Wallis test followed by Dunn's multiple comparisons test (**c**, **e**) or two-sided Wilcoxon test (**g**–**i**). Source data are provided as a Source Data file.

sensory stimulation-evoked axonal activity was also diminished in VTA−PL vGlutT2+ projections in mice with neuropathic pain (Supplementary Fig. 1c, d).

In a separate experiment, we assessed the somatic activity of VTA vGlutT2+ neurons before and after SNI. For this experiment, we injected Cre-dependent GCaMP6s into the VTA of *Vglut2*IRES-Cre mice and implanted a gradient-index (GRIN) lens above the VTA to enable in vivo Ca2+ imaging (Supplementary Fig. 2). Two weeks post-SNI, we observed a significant decrease in the overall activity of VTA vGluT2+ neurons compared to the pre-SNI baseline ($P = 0.0103$).

### Activation of VTA−PL glutamatergic inputs alleviates neuropathic pain-associated behaviors

To examine the potential contribution of reduced VTA−PL glutamatergic activity to neuropathic pain, we used an optogenetic approach to selectively activate these projections in mice subjected to SNI[27]. Specifically, we injected a Cre-dependent AAV expressing humanized channelrhodopsin (hChR2) fused with mCherry into the VTA of *Vglut2*IRES-Cre mice and then implanted a cannula guiding a 200 μm diameter fiber-optic above the PL to allow for light delivery (Fig. 2a). *Post hoc* immunohistochemistry analysis in the PL revealed a substantial increase in c-Fos+ cells following blue light delivery (Supplementary Fig. 3), confirming the activation of PL by optical stimulation of VTA glutamatergic terminals expressing hChR2.

We then assessed pain aversion in these mice using a real-time place preference (RT-PP) test (Fig. 2b)[28,29]. Two weeks after the SNI procedure, mice were placed in a three-compartment chamber, with one side designated for light stimulation, triggering laser delivery upon entry, while leaving the area turned the laser off. Remarkably, SNI mice expressing hChR2 exhibited a preference for staying in the chamber paired with light stimulation compared to those expressing mCherry without hChR2 (Fig. 2c), indicating the alleviation of aversion upon light activation of VTA−PL glutamatergic terminals. By contrast, sham mice expressing hChR2 showed no preference for light stimulation in the RT-PP test.

Similar findings were observed in a classic conditioned place preference (CPP) test (Fig. 2d)[30]. During the conditioning phase, one compartment was paired with light stimulation, while the other received no light. In the subsequent test phase, mice had unrestricted access to both compartments without light stimulation. SNI mice expressing hChR2 in VTA−PL glutamatergic neurons demonstrated a significant preference for the compartment associated with light compared to SNI mice expressing mCherry ($P = 0.0077$) or sham mice expressing hChR2 ($P = 0.0061$) (Fig. 2e), providing additional evidence for the analgesic effect achieved through the activation of VTA−PL glutamatergic terminals.

Furthermore, we assessed mechanical and thermal thresholds in SNI mice (Fig. 2f). Photoactivation of VTA glutamatergic inputs to the

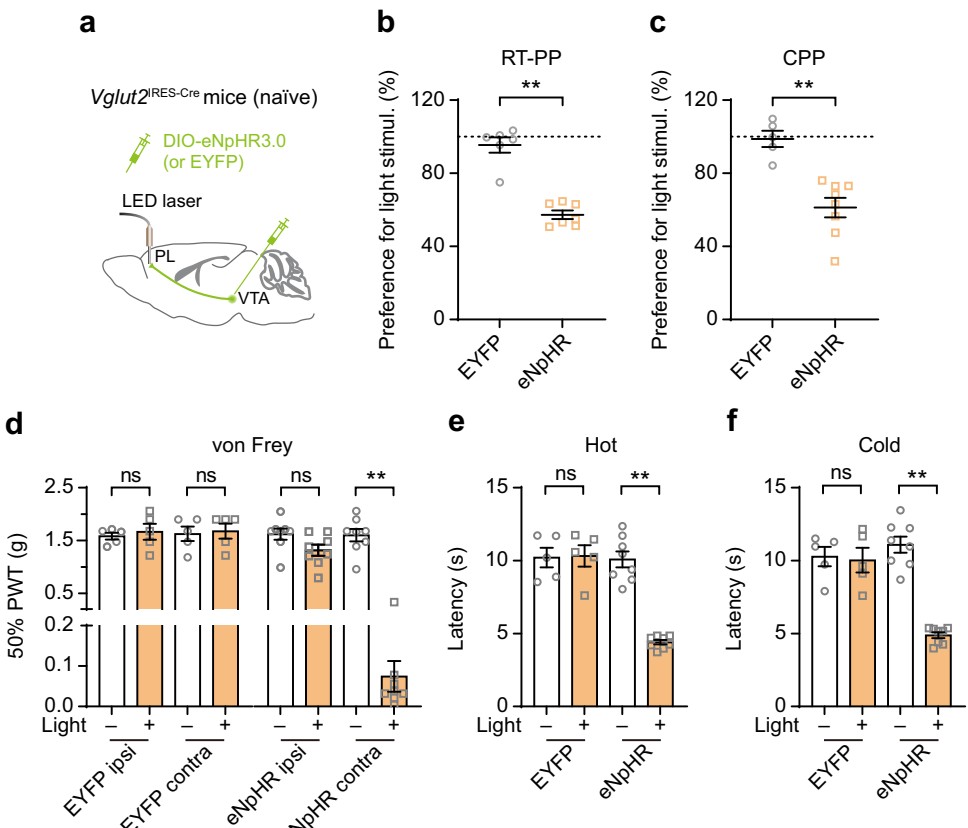

Fig. 3 | Inhibition of the VTA−PL glutamatergic pathway in naïve mice results in pain-like behaviors that resemble those observed in SNI mice. a Experimental design to express eNpHR3.0 or EYFP (control) in VTA glutamatergic neurons and stimulate their terminals in the PL of naïve mice. b Inhibiting VTA−PL glutamatergic terminals induces real-time aversion in naïve mice ($P = 0.0012$, n = 6, 7 mice). c Preference to stay in the light stimulation-paired chamber for naïve mice expressing EYFP or eNpHR3.0 in VTA glutamatergic neurons ($P = 0.0016$).

d−f Inhibiting VTA−PL glutamatergic terminals reduces the animal's mechanical ($P = 0.0078$), hot ($P = 0.0078$), and cold ($P = 0.0078$) thresholds in the limb contralateral to the viral injection side. In (c−f), n = 5, 8 mice for EYFP, eNpHR3.0 respectively. Summary data are presented as mean ± S.E.M. **$P < 0.01$; ns not significant; by Mann–Whitney test for unpaired comparison or Wilcoxon test for paired comparison, two-sided. Source data are provided as a Source Data file.

PL significantly increased the mechanical paw withdrawal thresholds in the nerve-injured limb ($P = 0.0078$), with no significant changes observed in the contralateral non-injured limb ($P = 0.1562$) (Fig. 2g). In contrast, light stimulation in SNI mice expressing mCherry without hChR2 did not produce any notable effects on the animals' mechanical thresholds (Fig. 2g). Moreover, the activation of VTA−PL glutamatergic terminals significantly increased the animals' thermal thresholds on both hot and cold plates (Fig. 2h, i). Together, these results demonstrate that the activation of the VTA−PL glutamatergic pathway effectively alleviates aversion and allodynia in mice with neuropathic pain.

### Inhibition of VTA−PL glutamatergic inputs in naïve mice elicits pain-like behaviors
To further determine the role of the VTA−PL glutamatergic pathway in pain modulation, we employed an inhibitory approach using a light-activated chloride channel in naïve mice[31]. Specifically, we administered a Cre-dependent AAV encoding halorhodopsin fused with an enhanced yellow fluorescent protein (eNpHR3.0) into the VTA of *Vglut2*[IRES-Cre] mice and implanted a fiber-optic above the ipsilateral PL for light delivery (Fig. 3a). Two weeks post-surgery, we performed optogenetic inhibition in both the RT-PP and CPP tests. Significantly, mice expressing eNpHR3.0 exhibited avoidance of the light-paired compartment compared to mice expressing EYFP (Fig. 3b, c), indicating that the inhibition of VTA glutamatergic terminals in the PL induces emotional aversion in normal mice. Furthermore, we examined whether optogenetic inhibition of VTA glutamatergic terminals in the PL

affected nociceptive thresholds in naïve mice. Indeed, inhibiting VTA−PL glutamatergic terminals in normal mice led to a decrease in mechanical and thermal thresholds in the paw contralateral to the brain region subjected to light stimulation (Fig. 3d−f). These results indicate that the inhibition of the VTA−PL glutamatergic pathway elicits pain-like behaviors in naïve mice without nerve injury.

### Activation of VTA−PL glutamatergic inputs alleviates pain independent of dopamine co-release
Approximately 40% of VTA glutamatergic neurons projecting to the PL co-express TH, an enzyme involved in dopamine synthesis[14,15,32]. This raises the possibility that the pain modulatory effect of VTA−PL glutamatergic projections may rely on their ability to recruit dopamine release. To test this, we used a recently developed viral-based CRISPR/Cas9 approach to selectively disrupt dopamine synthesis in PL-projecting VTA glutamatergic neurons. This approach utilized a single AAV vector for Cre-dependent expression of *Staphylococcus aureus* Cas9 (SaCas9) and U6 promoter-driven expression of a single-guide RNA to induce indel mutations in the TH gene (sg*Th*)[33]. We injected this AAV vector (or control vector sgCTRL) into the VTA of *Vglut2*[IRES-FLPo] mice and simultaneously administered a retrograde FLP-dependent Cre into the PL (Fig. 4a). Immunohistochemical analysis revealed that in the sgCTRL group, 43.51% of Cas9-expressing (vGluT2+) VTA−PL neurons expressed TH, whereas in the sg*Th* group, only 2.76% expressed TH (Fig. 4b, c). These results confirm the successful deletion of TH in VTA glutamatergic neurons projecting to the PL.

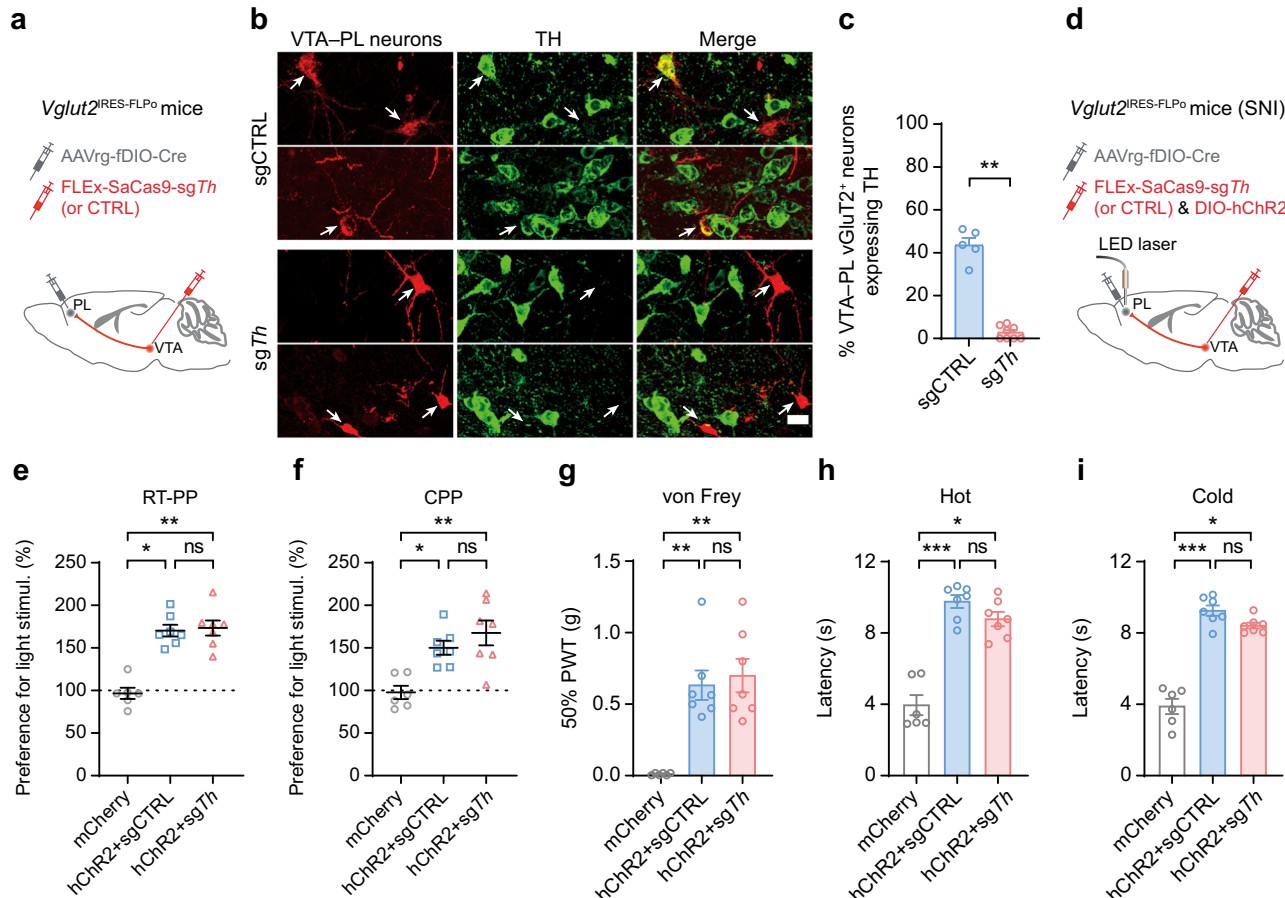

**Fig. 4 | VTA–PL glutamatergic inputs modulate pain-associated behaviors independent of dopamine signaling. a** Experimental design to express Cas9-sg*Th* or control (CTRL) in VTA–PL glutamatergic neurons. SaCas9 is fused with an HA tag on the C-terminus. **b** Confocal images of VTA showing colocalization of HA-tagged PL-projecting glutamatergic neurons and TH⁺ neurons. Scale bar, 20 μm. **c** Percentage of VTA–PL glutamatergic neurons expressing TH ($P = 0.0016$, $n = 5, 8$ sections). **d** Experimental design to express Cas9-sg*Th* (or CTRL) and hChR2 in VTA–PL glutamatergic neurons and stimulate their terminals in the PL of SNI mice. **e-f** Activating VTA–PL glutamatergic terminals induces real-time (**e**) and conditioned (**f**) place preference in SNI mice expressing hChR2 with sg*Th* (**e**, $P = 0.0050$; **f**, $P = 0.0052$) or CTRL (**e**, $P = 0.0102$; **f**, $P = 0.0238$). There is no difference between

sg*Th* and CTRL groups (**e**, **f**, $P > 0.9999$). **g–i** Activating VTA–PL glutamatergic terminals increases mechanical (**g**), hot (**h**), and cold (**i**) thresholds in SNI mice expressing hChR2 with sg*Th* (**g–i**, $P = 0.0055, 0.0426, 0.0480$) or CTRL (**g–i**, $P = 0.0085, 0.0009, 0.0007$). There is no difference between sg*Th* and CTRL group (**g–i**, $P > 0.9999, = 0.6677, 0.5705$). In (**e–i**), $n = 6, 7, 7$ mice for mCherry, hChR2+sgCTRL, hChR2+sg*Th*, respectively; Mice in the mCherry group are the same mice shown in Fig. 2. Summary data are presented as mean ± S.E.M. *$P < 0.05$, **$P < 0.01$, ***$P < 0.001$; ns not significant; by two-sided Mann–Whitney test or Kruskal–Wallis test followed by Dunn's multiple comparisons test. Source data are provided as a Source Data file.

To examine whether the activation of VTA–PL glutamatergic inputs, without simultaneous dopamine co-release, could alleviate neuropathic pain-associated behaviors in SNI mice, we combined the CRISPR/Cas9 approach with optogenetics (Fig. 4d). In the absence of dopamine, optogenetic activation of VTA glutamatergic terminals in the PL reduced aversion and allodynia (Fig. 4e-i), similar to the SNI group with intact dopamine signaling (Fig. 2). To exclude the potential effects of residual dopamine, even though it constituted a minor percentage (<3%), we employed pharmacological methods to locally block dopamine transmission during the photoactivation of VTA–PL glutamatergic terminals in SNI mice (Supplementary Fig. 4a). In the presence of D1 and D2 receptor antagonists (SCH23390 and sulpiride), the activation of VTA glutamatergic terminals in the PL significantly decreased aversion and allodynia in mice with neuropathic pain (Supplementary Fig. 4b, c). Together, these results demonstrate that the activation of the VTA–PL glutamatergic pathway is sufficient to alleviate neuropathic pain independent of concurrent dopamine co-release, suggesting the involvement of a non-dopamine mechanism by which the mesocortical pathway modulates pain.

## VTA neurons primarily project to excitatory neurons in the PL

Furthermore, we characterized the anatomical connectivity in the VTA–PL circuit using a virus-mediated transsynaptic tracing approach (Fig. 5)[34]. Specifically, we injected AAV1-FLPo and FLP-dependent mCherry into the anterior VTA and PL, respectively (Fig. 5a, c), enabling selective expression of mCherry in PL neurons that receive presynaptic inputs from the VTA (PL_VTA–PL neurons) through the anterograde transsynaptic spread capabilities of AAV serotype 1 (AAV1). Our results showed that the majority (~85%) of mCherry-labeled PL_VTA–PL cells were located in the deep layers (>300 μm subpial) of PL (Supplementary Fig. 5).

To further characterize these PL_VTA–PL neurons, we injected CaMKIIα-EGFP into the PL of C57BL/6 mice to label excitatory neurons, or used Cre-dependent EGFP in *Gad2*^IRES-Cre, *Pvalb*^T2A-Cre, *Sst*^IRES-Cre, or *Vip*^IRES-Cre mice to label specific subsets of inhibitory interneurons (Fig. 5a, c). Following three weeks of viral expression, confocal imaging of brain sections revealed mCherry-labeled PL_VTA–PL neurons and EGFP-labeled CaMKIIα⁺, GABAergic (GAD2⁺), parvalbumin-expressing (PV⁺), somatostatin-expressing (SST⁺), or vasoactive intestinal peptide-expressing (VIP⁺) neurons (Fig. 5b, d). Approximately 75%, 23%, 15%,

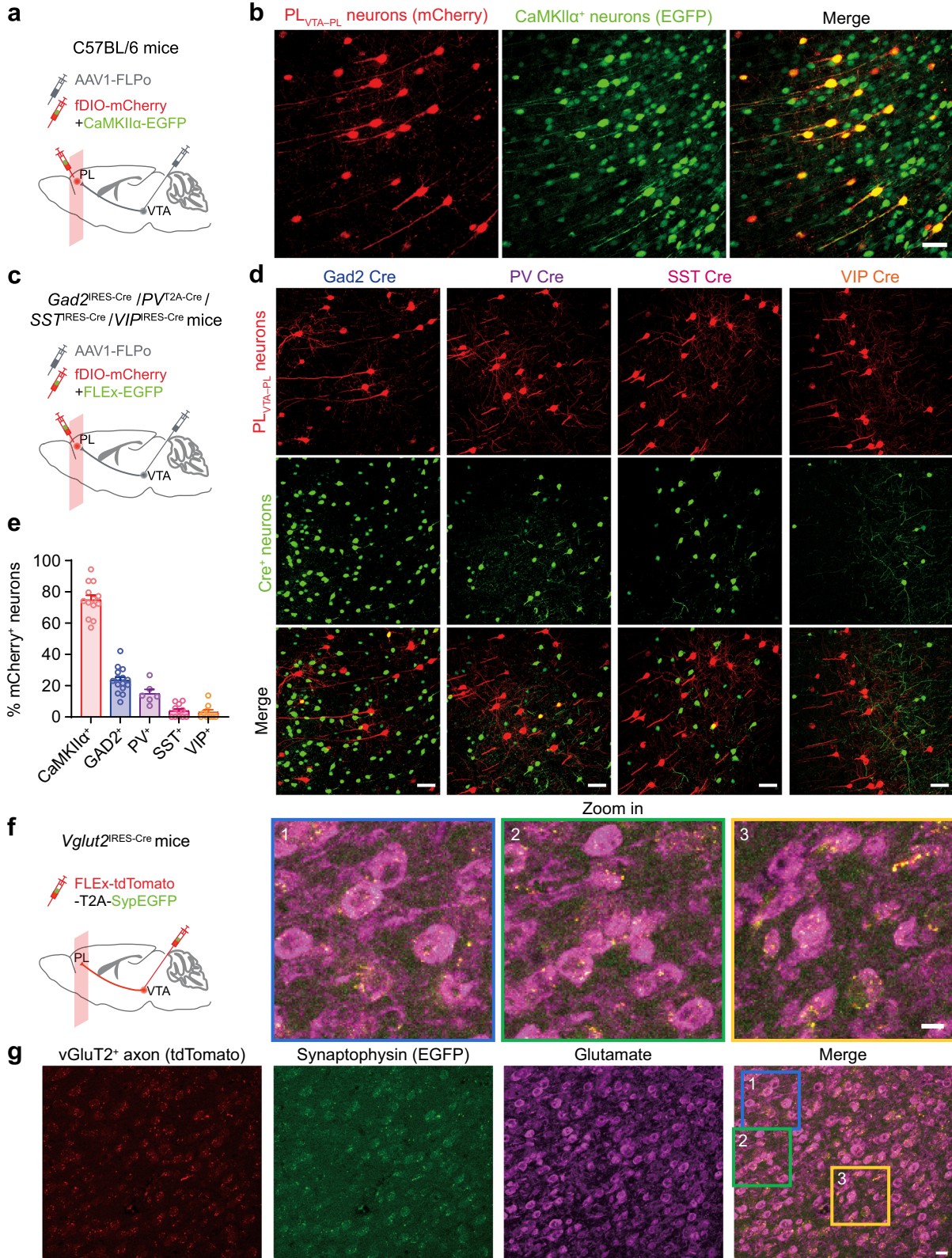

**Fig. 5 | Characterization of PL neurons that receive VTA inputs. a** Experimental design for expressing mCherry in PL neurons receiving VTA projections (PL$_{VTA-PL}$ neurons) and EGFP in CaMKIIα$^+$ neurons in the PL. **b** Confocal images of PL showing PL$_{VTA-PL}$ neurons and CaMKIIα$^+$ neurons. Scale bar, 50 μm. **c** Experimental design to express mCherry in PL$_{VTA-PL}$ neurons and EGFP in GAD2$^+$, PV$^+$, SST$^+$, or VIP$^+$ neurons in the PL. **d** Images showing PL$_{VTA-PL}$ neurons and GAD2$^+$/PV$^+$/SST$^+$/VIP$^+$ neurons. Scale bar, 50 μm. **e** Percentages of PL$_{VTA-PL}$

neurons expressing CaMKIIα (~75%), GAD2 (~23%), PV (~15%), SST (~4%), or VIP (~3%). Summary data are mean ± S.E.M. $n$ = 13, 15, 6, 11, 9 sections. **f** Experimental design to virally express presynaptic (synaptophysin-fused) EGFP in VTA glutamatergic neurons and image their terminals in the PL. **g** Confocal images showing VTA glutamatergic axons, presynaptic puncta (synaptophysin), and glutamate immunostaining in the PL (4 mice). Scale bar, 20 μm and 10 μm (zoom in). Source data are provided as a Source Data file.

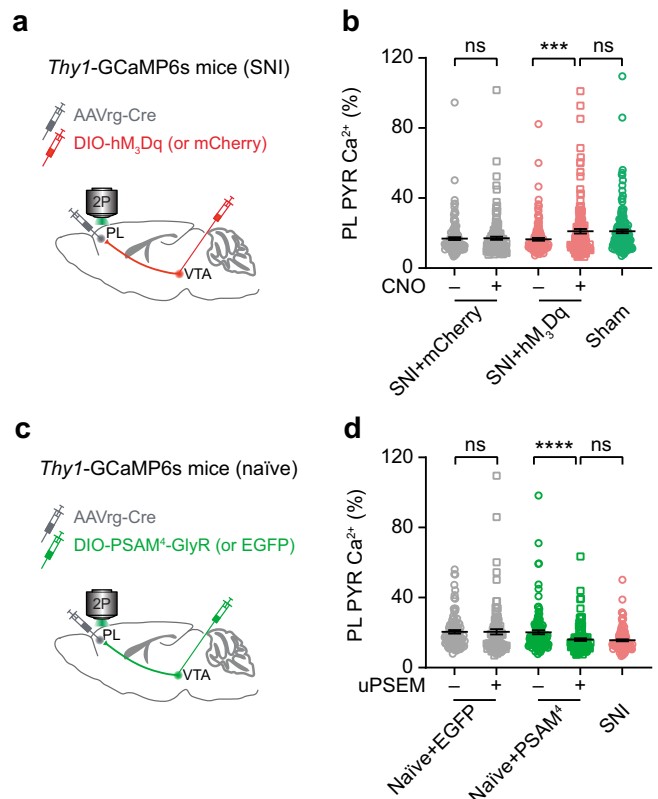

**Fig. 6 | Manipulation of VTA−PL projection neurons regulates pyramidal neuron activity in the PL. a** Experimental design for expressing hM₃Dq or mCherry (control) in VTA−PL projection neurons and in vivo Ca²⁺ imaging in PL pyramidal neurons. **b** Average Ca²⁺ activity in PL pyramidal neurons (PYR) before and after CNO application in SNI mice expressing mCherry ($P = 0.4220$; $n = 146$ cells from three mice) or hM₃Dq ($P = 0.0003$; $n = 142$ cells from three mice). Activation of VTA−PL projection neurons in SNI mice corrects PL pyramidal neuron activity to the level of sham mice ($n = 159$ cells from three mice; $P = 0.1286$). **c** Experimental design for expressing PSAM⁴-GlyR or EGFP (control) in VTA−PL projection neurons and in vivo Ca²⁺ imaging in PL pyramidal neurons. **d** Average Ca²⁺ activity in PL pyramidal neurons before and after uPSEM817 application in mice expressing EGFP ($P = 0.0959$; $n = 100$ cells from three mice) or PSAM⁴-GlyR ($P < 0.0001$; $n = 114$ cells from three mice). Inhibition of VTA−PL projection neurons in naïve mice reduces PL neuronal activity to levels comparable with SNI mice ($n = 122$ cells from three mice; $P = 0.7077$). Summary data are presented as mean ± S.E.M. ***$P < 0.001$, ****$P < 0.0001$; ns not significant; by Mann−Whitney test for unpaired comparison or Wilcoxon test for paired comparison, two-sided. Source data are provided as a Source Data file.

4%, and 3% of PL$_{VTA−PL}$ neurons were identified as CaMKIIα⁺, GAD2⁺, PV⁺, SST⁺, and VIP⁺ cells, respectively (Fig. 5e), indicating that neurons in the anterior VTA, which are mostly glutamatergic (Supplementary Fig. 6), primarily project to excitatory neurons in the PL. Consistent with these findings, we observed co-localization of VTA vGlutT2⁺ terminals with glutamate⁺ neurons in the PL (Fig. 5f, g).

### Activation of VTA−PL projections restores PL outputs to the ACC

Previous studies have shown that neuropathic pain leads to reduced activity of excitatory neurons in the PL[19,35]. To determine whether the reduction of VTA inputs contributes to PL hypoactivity, we used a chemogenetic approach to enhance the activity of PL-projecting VTA neurons in vivo (Fig. 6a)[36]. We found that activation of VTA−PL projection neurons caused a significant increase in pyramidal neuron activity in the PL of SNI mice, reaching levels comparable to those of sham mice ($P = 0.0003$ *vs*. CNO⁻, $P = 0.1286$ *vs*. sham) (Fig. 6b).

Consistently, chemogenetic activation of VTA−PL projection neurons attenuated neuropathic pain-associated aversion and allodynia (Supplementary Fig. 7a–d). In a separate set of mice, we used chemogenetics to suppress the activity of VTA−PL projections in naïve mice (Fig. 6c)[37]. As expected, silencing VTA−PL projections suppressed the activity of pyramidal neurons in the PL and induced pain-like behaviors in normal mice, resembling that observed in SNI mice (Fig. 6d, Supplementary Fig. 7e–h), indicating a significant contribution of VTA inputs to PL excitation and their involvement in pain modulation. These data suggest that VTA neurons primarily innervate excitatory neurons in the PL, providing direct excitation from the midbrain to the cortex, thereby regulating pain signals.

The ACC plays an important role in processing the emotional aspects of pain[38,39]. Previous studies have shown that the PL primarily sends excitatory projections to the ACC, which are compromised in neuropathic pain conditions[19]. Building upon our finding that excitatory neurons in the PL receive inputs from the VTA (Fig. 5), we examined whether these PL$_{VTA−PL}$ neurons project to the ACC. To do this, we selectively expressed axonal GCaMP6 in PL$_{VTA−PL}$ neurons (Fig. 7a). Two weeks post-surgery, in vivo Ca²⁺ imaging in the ACC revealed numerous axons expressing GCaMP (Supplementary Fig. 8). Notably, in mice with SNI, the axonal Ca²⁺ activity of PL$_{VTA−PL}$ neurons projecting to the ACC was approximately half of that observed in sham mice ($15.42 \pm 0.85\%$ *vs*. $29.33 \pm 1.86\%$, $P < 0.0001$; Fig. 7b). Importantly, chemogenetic activation of VTA neurons projecting to the PL significantly increased the axonal activity of PL neurons in the ACC ($P < 0.0001$), reaching levels comparable to those observed in sham mice ($P = 0.2246$; Fig. 7c, d). It is worth noting that the ACC also receives direct glutamatergic inputs from the VTA (Supplementary Fig. 9a, b). Unlike diminished glutamatergic inputs from the VTA to the PL in neuropathic pain conditions (Fig. 1), SNI mice exhibited no changes in VTA glutamatergic terminal activity in the ACC compared to sham mice (Supplementary Fig. 9b–d), suggesting a circuit-specific impact of neuropathic pain on VTA glutamatergic projections.

Finally, we examined the involvement of VTA−PL−ACC circuits in neuropathic pain behavior by optogenetically activating the axonal terminals of PL$_{VTA−PL}$ neurons in the ACC (Fig. 7e). Compared to SNI mice expressing mCherry, light stimulation induced a place preference in SNI mice expressing hChR2 (Fig. 7f, g), indicating relief from ongoing pain. Consistently, the activation of the axonal terminals of PL$_{VTA−PL}$ neurons in the ACC increased mechanical and thermal thresholds in SNI mice (Fig. 7h–j).

## Discussion

The malfunction of the mesocortical pathway has been implicated in pain disorders[10,40,41]. While extensive research has focused on dopamine transmission in this pathway, the specific role of glutamatergic signaling in pain modulation remains largely unknown. Here, we show that VTA glutamatergic activity is significantly reduced in the PL of mice with peripheral neuropathic pain. This decrease in VTA inputs contributes to the attenuated somatic activity of PL excitatory neurons and their axonal outputs to the ACC, while activating the VTA−PL−ACC pathway alleviates neuropathic pain-associated behaviors. Notably, this analgesic effect can be achieved by selectively activating VTA glutamatergic terminals in the PL without dopamine co-release. These findings highlight the important role of the mesocortical glutamatergic pathway in pain modulation and provide insights into the underlying neural circuits associated with neuropathic pain.

In addition to their role in reward and motivation, VTA circuits have been implicated in processing aversive experiences, primarily through the action of dopaminergic neurons projecting to various brain structures[42,43]. Previous studies have demonstrated dysregulation of dopamine release, alterations in dopamine receptor expression, and changes in the firing properties of dopaminergic neurons in the context of chronic pain[8,9,44]. VTA dopaminergic neurons project to

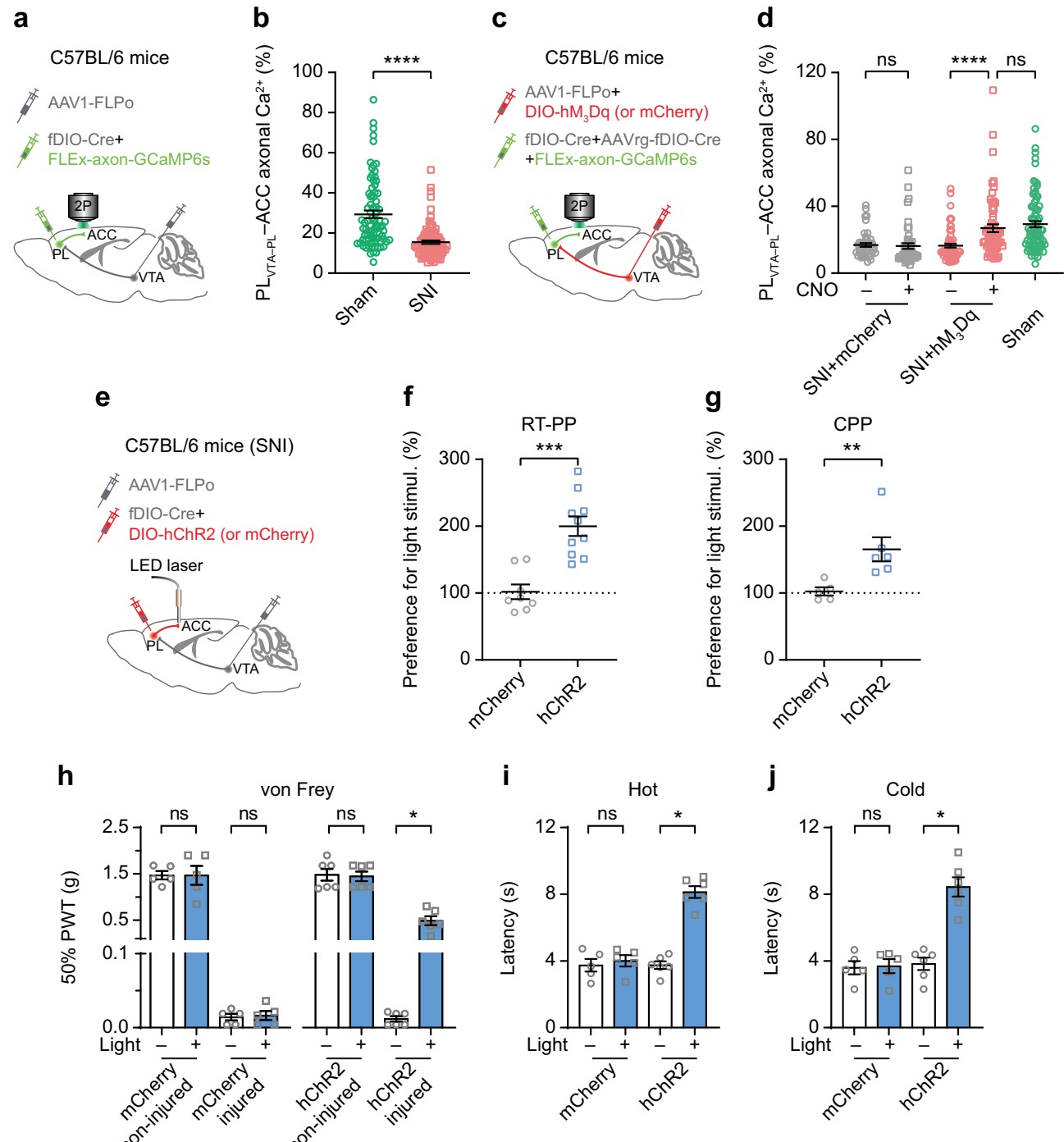

**Fig. 7 | Activation of VTA−PL projections restores PL outputs to the ACC and ameliorates pain-like behaviors. a** Experimental design to express axon-targeted GCaMP6s in $PL_{VTA−PL}$ neurons and in vivo $Ca^{2+}$ imaging of their terminals in the ACC. **b** ACC-projecting axonal $Ca^{2+}$ activity of $PL_{VTA−PL}$ neurons in sham and SNI mice ($P < 0.0001$; $n = 82/4$, 91/5 axon segments/mice). **c** Experimental design to express $hM_3Dq$ or mCherry in VTA−PL projection neurons and in vivo $Ca^{2+}$ imaging of $PL_{VTA−PL}$ terminals in the ACC. **d** ACC-projecting axonal $Ca^{2+}$ activity of $PL_{VTA−PL}$ neurons before and after CNO application in SNI mice expressing mCherry ($P = 0.1537$; $n = 48$ axon segments from four mice) or $hM_3Dq$ ($P < 0.0001$; $n = 62$ axon segments from four mice). Activation of VTA−PL projection neurons in SNI mice increases $PL_{VTA−PL}$ axonal activity to the level of sham mice ($n = 82$ axon segments from four mice; $P = 0.2246$). **e** Experimental design to express hChR2 or mCherry in $PL_{VTA−PL}$ neurons and stimulate their terminals in the ACC. **f** Activating $PL_{VTA−PL}$ terminals in the ACC induces real-time place preference in SNI mice ($P = 0.0002$; $n = 8$, 10 mice). **g** Preference to stay in the light stimulation-paired chamber for SNI mice expressing mCherry or hChR2 ($P = 0.0043$). **h–j**, Activation of $PL_{VTA−PL}$ terminals in the ACC increases the animals' mechanical ($P = 0.0312$), hot ($P = 0.0312$), and cold ($P = 0.0312$) thresholds in the limb ipsilateral to SNI. Light stimulation has no effect on mechanical and thermal sensitivity in SNI mice expressing mCherry. In (**g–j**), $n = 5$, 6 mice for mCherry, hChR2 respectively. Each dot represents data from an individual axon segment (**b**, **d**) or a single mouse (**f–j**). Summary data are mean ± S.E.M. *$P < 0.05$, **$P < 0.01$, ***$P < 0.001$, ****$P < 0.0001$; ns not significant; by Mann−Whitney test for unpaired comparison or Wilcoxon test for paired comparison, two-sided. Source data are provided as a Source Data file.

the PFC as part of the mesocortical pathway[2]. Activation of VTA dopaminergic inputs has been shown to enhance dopamine release in the PL and ameliorate behavioral deficits associated with neuropathic pain[10]. While significant research has focused on the involvement of dopaminergic neurons in pain modulation, around two-thirds of VTA neurons projecting to the PFC are glutamatergic[15]. VTA glutamatergic neurons are also known to play a role in reward and aversion[1]. However, the precise contribution of VTA–PL glutamatergic neurons in neuropathic pain remains largely unknown. Our study addresses this knowledge gap by revealing an unconventional mesocortical glutamatergic pathway in pain modulation: under chronic pain conditions, VTA–PL glutamatergic activity is suppressed, and enhancing VTA glutamatergic inputs to the PL alleviates pain-like behaviors in mice. These findings, in conjunction with previous reports[10], suggest a general reduction in excitatory inputs from the VTA to the PL in the presence of chronic pain.

A substantial fraction of VTA glutamatergic neurons has been found to release multiple neurotransmitters, including dopamine[15]. Under physiological conditions, the firing of VTA neurons in response to reward releases glutamate in the PFC, generating fast excitatory postsynaptic potentials for a rapid response to reward-related stimuli[45]. VTA neurons are thought to use this mechanism to convey temporally precise information to the PFC. In contrast, dopamine-mediated signaling has been shown to gate intrinsic inhibition in cortical neurons and modulate the excitability of PFC networks for prolonged periods[46,47]. Studies on the mesolimbic system suggest that the disruption of the co-release of dopamine and glutamate from VTA neurons reduces mice's responses to psychostimulants[48,49]. However, the extent to which the mesocortical glutamatergic pathway independently modulates pain or interacts with dopamine signaling remains unclear.

To address this, we employed a newly developed viral-based CRISPR/Cas9 approach to selectively deplete TH, thereby preventing the production and release of dopamine from VTA glutamatergic neurons projecting to the PL[33]. Our results demonstrate that the photoactivation of VTA glutamatergic terminals in the PL effectively alleviates aversion and allodynia, even in the absence of dopamine signaling. These findings highlight the existence of a distinct mesocortical pathway that can modulate neuropathic pain without relying on simultaneous dopamine release. This is consistent with a previous report indicating that the VTA glutamatergic pathway can drive positive reinforcement independent of dopamine signaling[17]. These studies suggest that VTA glutamatergic circuits involved in reward learning, emotional aversion, and touch sensation may function without the necessity of dopamine co-release. It is essential to note that our results do not preclude the role of dopamine signaling in pain modulation. Further studies are warranted to elucidate whether VTA glutamatergic and dopaminergic pathways function as parallel or overlapping systems in modulating pain sensation and affect.

The PFC is involved in top-down control of sensory and affective processes[50,51]. In animal pain models, previous studies have identified cellular dysfunctions within the PFC, including reduced excitatory glutamate release, abnormal cholinergic modulation, and decreased intrinsic excitability and firing rate of glutamatergic neurons[18,52–57]. Notably, the decline in excitatory inputs to the PFC has been implicated in chronic pain models[57]. Targeting local circuits within the PFC or its long-range inputs can enhance pyramidal neuron activity and alleviate pain-related behaviors in animals with chronic pain[10,19,35,56]. Consistent with these findings, our data show that the activation of VTA inputs, which preferentially target pyramidal neurons in the PL, effectively restores PL neuronal activity and alleviates aversion and allodynia in mice with neuropathic pain. Additionally, inhibiting VTA–PL inputs in naïve mice without peripheral nerve injury results in reduced PL pyramidal neuron activity and pain-like behaviors, thus supporting the causal role of VTA–PL projections in the development of pathological pain.

The PFC projects extensively to various brain regions implicated in pain processing, including the NAc, periaqueductal gray, and ACC[58–61]. Previous studies have shown that manipulating these PFC outputs, particularly those originating from the PL, can modulate chronic pain[19,62]. Under neuropathic pain conditions, we observed a substantial decrease in axonal outputs to the ACC from PL neurons that receive VTA projections. Remarkably, optogenetic activation of the VTA–PL–ACC pathway alone was sufficient to alleviate aversion and allodynia in mice with neuropathic pain. Through anterograde viral labeling, we further identified that the majority of PL$_{VTA-PL}$ neurons are CaMKIIα$^+$. Our previous study showed that nearly all ACC-projecting PL neurons are glutamatergic, and they mostly innervate GABAergic cells in the ACC[19]. Therefore, it can be inferred that in neuropathic pain, PL conveys less excitation from VTA glutamatergic to ACC GABAergic neurons, ultimately contributing to ACC disinhibition[63]. Importantly, chemogenetic activation of VTA–PL projection neurons effectively increased the PL outputs to the ACC, suggesting that activating VTA projections to the PL may ameliorate ACC dysfunction associated with neuropathic pain.

In summary, our study reveals the function of the VTA–PL glutamatergic pathway in modulating neuropathic pain, operating independently of dopamine signaling. Targeting mesocortical glutamate transmission shows promise as a potential therapeutic strategy for pain treatment.

## Methods

### Animals

*Vglut2*$^{IRES-Cre}$ (Jackson Laboratory, 016963), *Vglut2*$^{IRES-FLPo}$ (030212), *Gad2*$^{IRES-Cre}$ (010802), *Pvalb*$^{T2A-Cre}$ (012358), *Sst*$^{IRES-Cre}$ (013044), *Vip*$^{IRES-Cre}$ (010908), and C57BL/6 J (000664) mice were acquired from the Jackson Laboratory. *Thy1.2*-GCaMP6s transgenic mice (founder line 3), expressing GCaMP6s in cortical pyramidal neurons[64], were bred in-house. All mice were group-housed in temperature- and humidity-controlled rooms with a 12-h light-dark cycle. Male and female mice aged two to three months were used for all the experiments. All animal procedures were performed in accordance with protocols approved by the Institutional Animal Care and Use Committee at Columbia University as consistent with the National Institutes of Health (NIH) Guidelines for the Care and Use of Laboratory Animals. The experimenters were blinded to treatment groups.

### Spared nerve injury

Spared nerve injury (SNI) was conducted under sterile conditions[65,66]. Mice were anesthetized with an intraperitoneal (i.p.) injection of 100 mg/kg ketamine and 15 mg/kg xylazine. An incision was made in the thigh contralateral to the viral injection site, exposing the sciatic nerve. The tibial and common peroneal branches of the sciatic nerve were ligated and transected, while the sural nerve was left intact. Muscle and skin were closed and sutured in two layers. For the sham surgery, the sciatic nerve was exposed but not manipulated. Throughout the surgical procedure and recovery, the animal's body temperature was maintained at approximately 37 °C.

### Surgical preparation for imaging awake, head-restrained mice

To prepare a cranial window for in vivo Ca$^{2+}$ imaging in the PL or ACC[24,63], mice were deeply anesthetized with an i.p. injection of 100 mg/kg ketamine and 15 mg/kg xylazine. Following a scalp incision, a head plate (CF-10, Narishige) was attached to the animal's skull with glue and dental cement. A small section (~1 mm in diameter) of the skull over the PL (anterior-posterior (AP) + 2.68 mm, medial-lateral (ML) 0.5 mm) or ACC (AP + 0.75 mm, ML 0.5 mm) was carefully removed, without damaging the dura matter. A round glass coverslip, approximately the same size as the removed bone, was affixed to the skull using adhesive. Dental cement was applied to the surrounding area to further secure the glass window. Throughout the surgical

procedure and recovery, the animal's body temperature was maintained at approximately 37 °C. Imaging experiments were conducted 24 h after window implantation, free from anesthetic effects. Before imaging, mice were habituated three times (10 min each time) in the imaging platform to reduce potential stress associated with head restraining and imaging.

### In vivo Ca²⁺ imaging and data analysis
The genetically encoded Ca²⁺ indicator GCaMP6s was used for in vivo Ca²⁺ imaging. *Thy1*-GCaMP6s mice were used for Ca²⁺ imaging of pyramidal neurons in the PL. To image Ca²⁺ in the axons of VTA glutamatergic neurons, 0.2 μl of AAV5-hSynapsin1-FLEx-axon-GCaMP6s (112010; Addgene) was slowly injected into the VTA (AP −3.28 mm, ML 0.36 mm, subpial (SP) 4.13 mm) of *Vglut2*^IRES-Cre mice over 15 min using a picospritzer (15 p.s.i., 10 ms pulse width, 0.5 Hz) via a glass microelectrode. To image the axons of PL neurons that receive VTA projections (PL$_{VTA-PL}$ neurons), 0.1 μl of AAV1-EF1a-FLPo (55637; Addgene) was injected into the anterior VTA of C57BL/6 mice, allowing the virus to transduce anterogradely to the postsynaptic cells to express FLP. Concurrently, a 0.1 μl mixture of FLP-dependent Cre (AAV9-EF1a-fDIO-Cre; 121675; Addgene) and AAV5-hSynapsin1-FLEx-axon-GCaMP6s (volume 1:1) was injected into the PL (AP + 2.68 mm, ML 0.5 mm, SP 0.85 mm). Two to four weeks after viral injection, a cranial window was prepared for two-photon imaging. To image glutamatergic neurons in the VTA, 0.2 μl of AAV9-Syn-FLEx-GCaMP6s (100845; Addgene) was injected into the anterior VTA of *Vglut2*^IRES-Cre mice, followed by the implantation of a GRIN lens of 0.5 mm in diameter and 6.049 mm in length (CLHS050GFT039; Go!Foton) above the VTA[67]. Two to four weeks after surgery, animals were subjected to in vivo imaging.

In vivo two-photon imaging was performed using a Scientifica two-photon system equipped with a Ti:Sapphire laser (Vision S, Coherent) tuned to 920 nm. All experiments were performed using a 25× objective (1.05 N.A.) immersed in artificial cerebrospinal fluid (ACSF), with a digital zoom of 1× for somas and 3× for axons. All images were acquired at a frame rate of ~1.69 Hz (2-μs pixel dwell time) at a resolution of 512 × 512 pixels. Image acquisition was carried out using ScanImage software. Imaging with excessive movement was removed from the analysis.

Imaging data were analyzed using NIH ImageJ software. Regions of interest (ROIs) corresponding to visually identifiable somas and axons were selected for quantification[68]. The fluorescence time course of each ROI was determined by averaging all pixels within the ROIs. GCaMP6 fluorescence values were first adjusted by subtracting a background value. All Ca²⁺ transients were calculated as $\Delta F/F_0$, where $\Delta F/F_0$ is $(F-F_0)/F_0$, and $F_0$ represents the baseline value defined as the fluorescence averaged over the 6-s lowest period of fluorescence signal during the recording. Ca²⁺ activity was quantified as the average of $\Delta F/F_0$ over the recording period.

### Optogenetic manipulation
To activate VTA glutamatergic terminals in the PL, 0.2 μl of Cre-dependent hChR2 (AAV9-EF1a-double floxed-hChR2(H134R)-mCherry; 20297; Addgene) or mCherry (control) (AAV9-hSyn-DIO-mCherry; 50459; Addgene) was injected into the VTA of *Vglut2*^IRES-Cre mice.

For activation of axons of PL neurons receiving VTA projections, 0.1 μl of AAV1-EF1a-FLPo (55637; Addgene) was injected into the anterior VTA of C57BL/6 mice, and a 0.2 μl mixture (volume 1:1) of AAV9-EF1a-fDIO-Cre and AAV9-EF1a-double floxed-hChR2(H134R)-mCherry or AAV9-hSyn-DIO-mCherry (control) was injected into the PL.

To inhibit VTA glutamatergic terminals in the PL, 0.2 μl of AAV9-Ef1a-DIO-eNpHR3.0-EYFP (26966; Addgene) or AAV9-Ef1a-DIO-EYFP (control) (27056; Addgene) was injected into the VTA of *Vglut2*^IRES-Cre mice.

A custom-made fiber-optic (Ø 200 μm; FG200LEA; ThorLabs) with a ceramic ferrule (Ø 1.25 mm; CFLC126-10; ThorLabs) was implanted above the designated brain regions under stereotaxic guidance to deliver light stimulation. Blue light (470 nm, ~1 mW) generated by a fiber-coupled LED (M470F3; ThorLabs) triggered by a TTL pulse generator (OTPG-4; Doric Lenses) at 20 Hz with a 20-ms pulse width was used to activate hChR2. Continuous yellow light (595 nm, 1–2 mW) generated by a fiber-coupled LED (M595F2; ThorLabs) was used to drive eNpHR3.0.

### Chemogenetic manipulation
To activate VTA–PL projection neurons in *Thy1*-GCaMP6s mice, 0.2 μl of retrograde AAV encoding Cre (AAVrg-Pgk-Cre; 24593; Addgene) was injected into the PL, and the ipsilateral VTA was injected with 0.2 μl of AAV9-hSyn-DIO-hM₃D(Gq)-mCherry (44361; Addgene) or AAV9-hSyn-DIO-mCherry (control). After SNI surgery on the hindlimb contralateral to the AAV injection site, mice expressing hM₃Dq or mCherry were given 5 mg/kg of clozapine *N*-oxide (CNO, 1 mg/ml in saline, i.p.). In vivo Ca²⁺ imaging was performed before and 30 min after CNO administration.

To inhibit VTA–PL projection neurons, 0.2 μl of AAVrg-Pgk-Cre was injected into the PL, and the ipsilateral VTA was injected with 0.2 μl of Cre-dependent PSAM⁴-GlyR (AAV5-SYN-FLEx-PSAM4-GlyR-EGFP; 119741; Addgene) or EGFP (AAV1-CAG-FLEx-EGFP; 51502; Addgene). After 2–3 weeks, mice expressing PSAM4-GlyR or EGFP (control) were administered 0.3 mg/kg uPSEM817 (6866; Tocris) by i.p. injection. In vivo Ca²⁺ imaging was performed before and 30 min after uPSEM817 injection.

### Cell type-specific gene mutagenesis
A single AAV vector containing a recombinase-dependent *Staphylococcus aureus* Cas9 (SaCas9) and a single guide RNA (sgRNA) was used for cell type-specific gene mutagenesis[33]. Plasmids of CMV-FLEx-SaCas9-U6-sg*Th* (159901; Addgene) and FLEX-SaCas9-U6-sgRNA (control) (124844; Addgene) were packaged into AAV1 by Virovek Inc (~2E + 13 vg/mL). To selectively deplete tyrosine hydroxylase (TH) from VTA–PL glutamatergic neurons, 0.2 μl of retrograde AAV encoding FLP-dependent Cre (AAVrg-EF1a-fDIO-Cre; 121675; Addgene) was injected into the PL of *Vglut2*^IRES-FLPo mice to enable projection- and cell type-specific expression of Cre. Simultaneously, 0.2 μl of AAV1-FLEx-SaCas9-U6-sg*Th* or AAV1-FLEx-SaCas9-U6-sgRNA was injected into the ipsilateral VTA.

### Pharmacological manipulation
The D1 receptor antagonist SCH23390 hydrochloride (50 ng; 0925; Tocris) and the D2 receptor antagonist (s)-(-)-sulpiride (10 ng; 0895; Tocris) were dissolved in DMSO and then diluted into 0.2 μl of ACSF. An equal volume of vehicle without antagonists was used as a control. Using a glass microelectrode, 0.2 μl of the antagonists or vehicle was slowly injected (Picospritzer III; 15 p.s.i., 10 ms, 0.5 Hz) into the PL of SNI mice expressing hChR2 over 15 min. Behavior tests were conducted before and 10 min after the application of antagonists.

### Behavioral tests
**Real-time place preference (RT-PP).** The testing apparatus consisted of a custom-made chamber divided into three compartments. The side compartment measured 25 cm × 20 cm × 30 cm, while the middle compartment measured 10 cm × 10 cm × 30 cm. Each compartment had unique visual and textured cues, including distinct wall and floor patterns and textures. The RT-PP test was conducted approximately 3 weeks after viral injection, which was around 2 weeks after SNI/sham surgery. During the pre-conditioning phase, mice were placed in the middle compartment with unrestricted access to both sides. In the testing phase, one side was designated for light stimulation, where entries triggered the activation of a laser, and leaving the side turned off the laser. Each session lasted 20 min, and the amount of time spent in each compartment was recorded using video tracking software (ANY-maze). Animals that exhibited a strong unconditioned

preference (*i.e.*, spending >75% of the time in one compartment) were excluded from the analysis. The preference change was calculated as the ratio of time spent in the light stimulation-paired compartment between the test and pre-conditioning phase, with the preference for the treatment side calculated as test / pre-conditioning × 100%.

**Conditioned place preference (CPP)**. The testing apparatus used was a two-compartment chamber (Ugo Basile; Italy), measuring 32 cm × 15 cm × 25 cm. The chamber consisted of two equal-sized compartments connected by a removable door (4 cm wide × 6 cm high). Each compartment featured distinct visual and textured cues, such as unique wall patterns, floor patterns, and textures. The CPP test was conducted approximately 3 weeks after viral injection, which was ~2 weeks after SNI/sham surgery. During the pre-conditioning phase on day 1, mice were given free access to both compartments with the door removed. Two sessions, each lasting 20 min, were recorded. The time spent by each mouse in each compartment was recorded. Animals displaying a strong uncondi-tioned preference (>75% of time in one compartment) were exclu-ded from the analysis. During the conditioning phase on days 2 and 3, mice were confined to one compartment for 30 min, and the optic fiber was connected without light stimulation. Six hours later, mice were placed in the opposite compartment, which was paired with blue or yellow light stimulation as described earlier, for 30 min. The compartment assignments were counterbalanced among all the test mice. The test was conducted on day 4, during which mice were given free access to both compartments of the CPP chamber for 20 min, and the time spent in each compartment was recorded for each animal. The preference change was calculated as the ratio of time spent in the light-paired compartment between the test and pre-conditioning phases. The preference for the treatment side was determined as test / pre-conditioning × 100%.

**von Frey tests**
We used the up-and-down method with minor modifications to mea-sure the paw withdrawal thresholds of the animals[67,69]. Individually, mice were placed in clear acrylic boxes (10 cm × 7 cm × 7 cm) over a mesh table and allowed to habituate for 30 min before the testing procedure. A series of von Frey fibers (0.008, 0.02, 0.04, 0.07, 0.16, 0.4, 0.6, 1.0, 1.4, 2.0, and 4.0 g) were presented in consecutive ascending order. If there was no paw withdrawal response, the next stronger stimulus was applied. Conversely, if a paw withdrawal response occurred, the next weaker stimulus was chosen. Once the response threshold was crossed, six data points were collected. The first two responses of the series of six, which straddled the threshold, were retrospectively designated. The calculation of the 50% response threshold was performed as follows: 50% g threshold $= 10^{(X_f + \kappa\delta - 4)}$, where $X_f$ represents the value (in log units) of the final von Frey fiber used; $\kappa$ is the tabular value for the pattern of positive/negative responses; and $\delta$ is the mean difference (in log units) between stimuli (here, 0.2699)[70].

**Thermal tests**
Hot allodynia was assessed using a hot plate (Ugo Basile 7280; Italy). For cold allodynia testing, a custom-made cold plate was assembled using a thermoelectric Peltier cold plate (10 cm × 8 cm) (CP-031; TE Technologies, USA) and a temperature controller (TC-720; TE Tech-nologies, USA). Before testing, each mouse was placed individually in a clear acrylic container (10 cm × 7 cm × 20 cm) positioned on the plate at room temperature, allowing for a 30-min habituation period. During the testing phase, the temperature was set to 50 °C for the hot plate and 0 °C for the cold plate. Paw withdrawal response latency was recorded using a stopwatch, measuring the time it took for hind paw lifting coupled with flinching. Each animal's paw withdrawal latency was measured three times at 30-min intervals.

## Cell-type specific circuit dissection
To visualize the somas of VTA–PL glutamatergic neurons, 0.2 μ of AAVrg-EF1a-fDIO-Cre was injected into the PL, and 0.2 μl of AAV9-hSyn-DIO-mCherry was injected into the ipsilateral VTA of *Vglut2*[IRES-FLPo] mice.

For labeling the pre-synapses of VTA glutamatergic neurons, 0.2 μl of Cre-dependent synaptophysin-fused EGFP with tdTomato (AAV1-phSyn1(S)-FLEX-tdTomato-T2A-SypEGFP; 51509; Addgene) was injec-ted into the VTA of *Vglut2*[IRES-Cre] mice.

To label PL neurons that receive monosynaptic projections from VTA (PL$_{VTA-PL}$ neurons), 0.1 μl of AAV1-EF1a-FLPo (55637; Addgene) was injected into the anterior part of the VTA to enable the anterograde spread of the virus to the postsynaptic neurons for FLP expression. Simultaneously, a 0.2 μl mixture of FLP-dependent mCherry (AAV1-EF1a-fDIO-mCherry; 114471; Addgene) and AAV5-CaMKIIα-EGFP (50469; Addgene) (diluted 20×, 1:1) was injected into the PL of C57BL/6 mice to visualize PL$_{VTA-PL}$ neurons and CaMKIIα$^+$ neurons. To label GABAergic, PV$^+$, SST$^+$, or VIP$^+$ neurons, Cre-dependent EGFP (AAV1-CAG-FLEx-EGFP; 51502; Addgene) was injected into the PL of *Gad2*[IRES-Cre], *Pvalb*[T2A-Cre], *Sst*[IRES-Cre], or *Vip*[IRES-Cre] mice, respectively.

To visualize VTA–PL projection neurons, 0.2 μl of AAVrg-EF1a-FLPo (55637; Addgene) was injected into the PL of *Vglut2*[IRES-Cre] mice. Simultaneously, the ipsilateral VTA was injected with a 0.2 μl mix-ture of AAV1-EF1a-fDIO-mCherry and AAV1-CAG-FLEx-EGFP to characterize mCherry$^+$ PL projecting neurons and EGFP$^+$ glutama-tergic neurons. Mice were perfused three weeks after the AAV injection, and their brains were collected for subsequent experiments.

## Immunohistochemistry and confocal imaging
Mice were deeply anesthetized and transcardially perfused with a phosphate-buffered solution (PBS) followed by 4% paraformaldehyde (PFA). The brains were carefully removed and post-fixed in 4% PFA at 4 °C overnight. Subsequently, they were incubated in 20% and 30% (w/v) sucrose for 24 h each to ensure proper dehydration. Coronal sec-tions with a thickness of 20 or 30 μm were prepared using a cryostat (Microm HM505E).

For immunofluorescence, floating sections were permeabilized and blocked in 0.1% Triton X-100 with 10% donkey serum in PBS for 1 h at room temperature. They were then incubated overnight at 4 °C with primary antibodies: rabbit anti-c-Fos (1:500, 226008; Synaptic Systems), sheep anti-tyrosine hydroxylase (1:800, AB1542; Sigma Aldrich), rabbit anti-HA tag (1:500, ab9110; Abcam) (SaCas9 has a hemagglutinin (HA)-epitope tag on the C terminus), and rabbit anti-glutamate (1:500, G6642; Sigma Aldrich). After three washes with PBS, the sections were incubated for 2 h at room temperature with corresponding fluorophore-conjugated secondary antibodies as follows: donkey anti-sheep DyLight 405 (1:400, 713475003; Jackson Immuno Research Labs), donkey anti-sheep Alexa Fluor 647 (1:400, A21448; Invitrogen), donkey anti-rabbit CF543 (1:400; 20308, Biotium), donkey anti-rabbit Alexa Fluor 647 (1:400; A31573, Invitrogen). Following additional washing steps, the sections were mounted in medium (010001; SouthernBiotech) for confocal imaging.

Confocal imaging was performed using a Nikon Ti laser scanning confocal system with a 20× objective. Images were captured at 1024 × 1024 pixels with a resolution of 0.622 μm/pixel for visualizing somas and 0.207 μm/pixel for visualizing axonal terminals. Z-stacks of images (20 or 30 μm thick) were collected at 2.5-μm step sizes for somas and 0.7-μm for axons. The images were then projected at maximum intensity to create the final multi-channel images. ImageJ software was used for subsequent analysis.

## Statistics
All statistical analyses were performed using Prism 9.0 or 10.1.0 (GraphPad Software, Inc). Summary data were presented as

mean ± S.E.M. Sample sizes were selected to ensure sufficient statistical power while minimizing the number of animals used. Animals that were successfully measured were not excluded from the analysis. Data normality was assessed using the Shapiro–Wilk test, and nonparametric statistical tests were employed. The Mann–Whitney *U* test (or Wilcoxon matched-pairs signed rank test) was used to compare two unmatched (or matched) groups, and Dunn's multiple comparisons test was used for comparisons involving more than two groups. All comparisons were two-tailed. The significance level was set at $P < 0.05$. Detailed statistical information is provided in the figure legends.

### Reporting summary

Further information on research design is available in the Nature Portfolio Reporting Summary linked to this article.

## Data availability

The data supporting the findings of this study can be found in the paper and its supplementary information. Source data are provided with this paper.

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

## Acknowledgements

We thank Dr. Hang Zhou for technical assistance and input, as well as all the members of the Yang Lab for their helpful discussions. This work was supported by National Institutes of Health grants R01AA027108 (to GY).

## Author contributions

M.L. and G.Y. designed the studies. M.L. conducted the experiments and carried out data analysis. Both authors contributed to data interpretation. M.L. and G.Y. wrote the manuscript.

## Competing interests

The authors declare no competing interests.
