## [Peer Review File · Nature Communications]

A mesocortical glutamatergic pathway modulates neuropathic pain independent of dopamine co-releaseREVIEWER COMMENTS

Reviewer #1 (Remarks to the Author):

The study by Li and Yang used SNI-induced chronic neuropathic pain model in mice to demonstrate that VTA-PL glutamatergic activity is reduced and activation of VTA-PL glutamatergic inputs alleviates neuropathic pain-associated behaviors. Inhibition of VTA-PL glutamatergic inputs in naïve mice elicits pain-like behaviors. Furthermore, they found that above effect is independent of dopamine co-release. Finally, they demonstrated that the activation of VTA-PL projections restores somatic activity in PL neurons and enhances their axonal outputs to the anterior cingulate cortex, thus effectively ameliorating neuropathic pain-associated behaviors. The study has broad implications for understanding the functional role and mechanism of VTA glutamatergic neurons in pain regulation. The manuscript is carefully written, and the narrative is clear. I have a few concerns as follows:

1. Although not much study was reported, several following studies about involvement of VTA glutamatergic neurons in pain regulation should be mentioned in the INTRODUCTION.
 - 1) Root DH, et al. Role of glutamatergic projections from ventral tegmental area to lateral habenula in aversive conditioning. *J Neurosci*. 2014 Oct 15;34(42):13906-10.
 - 2) Abdul M, et al. VTA-NAc glutaminergic projection involves in the regulation of pain and pain-related anxiety. *Front Mol Neurosci*. 2022 Dec 7;15:1083671.
 - 3) Wang et al., Rewarding Effects of Optical Stimulation of Ventral Tegmental Area Glutamatergic Neurons. *J Neurosci*, 2015 Dec 2;35(48):15948-54.
2. VTA-PL glutamatergic basal activity is reduced in mice with neuropathic pain. How about its response to a thermal or mechanical stimulation?
3. How many percentage that VTA glutamatergic neurons projecting to PL has in all VTA glutamatergic neurons?
4. What are physiological functions for co-release of dopamine in PL terminals of VTA glutamatergic neurons? And, whether dopamine-release in PL modulates the release of glutamate?
5. The changes of overall VTA glutamatergic activity in mice with neuropathic pain should be measured?
6. Whether has functional monosynaptic connection between the VTA glutamatergic neurons and PL glutamatergic neurons should be tested with electrophysiological method?
7. Which receptor (NMDAR or AMPAR) in PL glutamatergic neurons mediates this effect of pain regulation?
8. Whether direct VTA glutamatergic neurons inverts into the ACC and its functional role in the development of chronic neuropathic pain?

Reviewer #2 (Remarks to the Author):

Li & Yang report in this manuscript that spared nerve injury (SNI) leads to a reduction in the activity of VTA-VGluT2+ terminals in the prelimbic cortex. They report using optogenetic and chemogenetic strategies that activation of this projection alleviates pain systems and increases the activity of glutamatergic PL neurons synaptically linked to VTA-VGluT2+ neurons, whereas inhibition of this circuit induces neuropathic pain-like behaviors in intact mice and reduced the activity of PL neurons. Using CRISPR/Cas9 deletion of tyrosine hydroxylase, they demonstrate that dopamine co-release is not involved in the alleviation of pain-like behaviors via activation of this circuit. Finally, they find that activation of VTA-VGluT2-PL leads to increased activity of PL terminals in the ACC and that optogenetic activation of PLVTA-PL terminals in the ACC similarly alleviates pain-like behaviors. Overall, this is a well-conducted series of experiments elucidating the role of glutamatergic signaling from the anterior VTA, a region where ~2/3 of neurons are VGluT2+, to the PL cortex in neuropathic pain.

There are some concerns with interpretation of certain experimental results and methods that need to be addressed before the manuscript is published.

1) The authors state that they used the rtPP and CPP assays to assess pain aversion (line 109) and interpret the induced preference in both assays to indicate pain relief in mice with SNI. This is puzzling given that the canonical interpretation of this type of results is that the manipulation under investigation is rewarding/reinforcing. How these results relate to pain relief is therefore unclear, especially since the authors do not report or cite previous literature showing the effects of these manipulations in intact mice.

2) The authors report that they conducted their optogenetic studies using 1mW (Blue) or 1-2mW (yellow) light stimulation. These are extremely low stimulation intensities for terminal modulation and inconsistent with the generally reported ~ 10 mW stimulation for optogenetic terminal stimulation. This concern would be alleviated by the author demonstrating that activation resulted in cFos expression in the PFC of the experimental cohorts.

Minor Comments:

1) In which cortical layer(s) were PLVTA-PL neurons located?

2) It is not completely clear what the authors mean by neuron deactivation. Does this imply that the neurons are simply less active generally? Clearly the activity of these neurons can be increased as shown in the results of Figure 7. Clarification of this term would be useful to prevent ambiguity in understanding the interpretation of these results.

Point-by-point responses to reviewers' comments

We thank the reviewers for their constructive comments on the manuscript. We have now comprehensively addressed the reviewers' concerns and made substantial revisions accordingly. Below we detail the changes and point-by-point responses to the comments.

Reviewer #1 (Remarks to the Author):

The study by Li and Yang used SNI-induced chronic neuropathic pain model in mice to demonstrate that VTA-PL glutamatergic activity is reduced and activation of VTA-PL glutamatergic inputs alleviates neuropathic pain-associated behaviors. Inhibition of VTA-PL glutamatergic inputs in naïve mice elicits pain-like behaviors. Furthermore, they found that above effect is independent of dopamine co-release. Finally, they demonstrated that the activation of VTA-PL projections restores somatic activity in PL neurons and enhances their axonal outputs to the anterior cingulate cortex, thus effectively ameliorating neuropathic pain-associated behaviors. The study has broad implications for understanding the functional role and mechanism of VTA glutamatergic neurons in pain regulation. The manuscript is carefully written, and the narrative is clear. I have a few concerns as follows:

We thank the reviewer for the positive and encouraging assessment of our work.

1. Although not much study was reported, several following studies about involvement of VTA glutamatergic neurons in pain regulation should be mentioned in the INTRODUCTION.
 - 1) Root DH, et al. Role of glutamatergic projections from ventral tegmental area to lateral habenula in aversive conditioning. *J Neurosci*. 2014 Oct 15;34(42):13906-10.
 - 2) Abdul M, et al. VTA-NAc glutaminergic projection involves in the regulation of pain and pain-related anxiety. *Front Mol Neurosci*. 2022 Dec 7;15:1083671.
 - 3) Wang et al., Rewarding Effects of Optical Stimulation of Ventral Tegmental Area Glutamatergic Neurons. *J Neurosci*, 2015 Dec 2;35(48):15948-54.

As suggested, we have now included these studies in the Introduction.

The relevant text in the Introduction now reads:

“Earlier studies have demonstrated that VTA glutamatergic neurons project to the nucleus accumbens (NAc) and lateral habenular, contributing to reward, aversion, and chronic pain^{1, 2, 3}. However, the specific role of the VTA-PFC glutamatergic pathway in pain modulation remains unclear.”

2. VTA-PL glutamatergic basal activity is reduced in mice with neuropathic pain. How about its response to a thermal or mechanical stimulation?

As suggested by the reviewer, we have conducted new experiments to measure evoked responses to mechanical stimulation in VTA-PL glutamatergic projections. Specifically, we applied punctate pressure stimuli to the animals' contralateral hind paw using a micropipette plunger (#5-000-1001-X10, Drummond) and performed *in vivo* Ca^{2+} imaging in VTA glutamatergic terminals in the PL. The Ca^{2+} activity averaged over 20 s after stimulation was $25.78 \pm 1.31\%$ in SNI mice ($n = 145$ axon segments from six mice) and $36.08 \pm 1.97\%$ in sham mice ($n = 172$ axon segments from six mice). Consistent with spontaneous activity, sensory stimulation-evoked axonal activity was also reduced in VTA-PL glutamatergic projections in mice with neuropathic pain ($P < 0.0001$; new Supplementary Fig. 1c, d).

New Supplementary Fig. 1: VTA-PL glutamatergic activity is decreased in both male and female mice with neuropathic pain.

a, SNI reduces spontaneous Ca^{2+} activity in VTA-PL glutamatergic axons in both males ($P < 0.0001$; $n = 116, 115$ axon segments from three mice per group) and females ($P < 0.0001$; $n = 139, 117$ axon segments from three mice per group). There is no significant difference between the sexes (sham, $P = 0.4516$; SNI, $P = 0.1345$). **b**, Distribution plot of data shown in **a**. **c**, Population average response of VTA glutamatergic terminals in the PL before and after

mechanical stimulation. Shading indicates S.E.M. **d**, Sensory stimulus-evoked Ca^{2+} activity in VTA-PL glutamatergic terminals is lower in SNI mice than sham mice ($P < 0.0001$; $n = 172$ and 145 axon segments from six mice per group). Each dot indicates data from a single axon segment (**a, d**). Summary data are presented as mean \pm S.E.M. **** $P < 0.0001$; by Mann-Whitney test (**a, d**).

3. How many percentage that VTA glutamatergic neurons projecting to PL has in all VTA glutamatergic neurons?

In the revised manuscript, we have quantified the percentage of VTA glutamatergic neurons projecting to PL among all VTA glutamatergic neurons. In this experiment, we injected a retrograde transducing adeno-associated virus (AAVrg) encoding FLP into the PL of *Vglut2*^{IRES-Cre} mice, along with AAV expressing FLP-dependent mCherry and Cre-dependent EGFP into the VTA (**new Supplementary Fig. 6a**). This injection strategy allowed for the selective expression of mCherry in VTA cells projecting to PL and EGFP in VTA glutamatergic cells, respectively. Two weeks after viral infection, we conducted confocal imaging in the VTA to characterize these cells. Our analysis showed that among all glutamatergic neurons in the VTA, approximately 5% of them project to the PL (**new Supplementary Fig. 6b, d**). These results align with previous reports that VTA glutamatergic neurons project to various brain regions, including ventral striatum^{2,4}, lateral habenular¹, ventral pallidum⁵, and amygdala, etc. However, in the anterior VTA, most (~75%) of PL-projecting neurons are glutamatergic (**Supplementary Fig. 6b, c**).

New Supplementary Fig. 6: The majority of neurons projecting to the PL in the anterior VTA are glutamergic.

a, Experimental design for expressing mCherry in VTA neurons projecting to PL (VTA-PL projection neurons) and EGFP in VTA glutamergic (vGluT2⁺) neurons. **b**, Confocal images of VTA showing colocalization of mCherry-labeled VTA-PL projection neurons and EGFP-labeled VTA vGluT2⁺ neurons. Arrows indicate VTA-PL vGluT2⁺ neurons; Arrowheads indicate VTA-PL vGluT2⁻ neurons. Scale bar, 20 μm. **c**, In the anterior VTA (AP -2.92 ~ -3.40 mm), most of the PL-projecting neurons are glutamergic (*n* = 8 mice). **d**, Percentage of PL-projecting glutamergic neurons among all glutamergic neurons in the VTA (*n* = 20 slices from four mice). Summary data are presented as mean ± S.E.M.

4. What are physiological functions for co-release of dopamine in PL terminals of VTA glutamergic neurons? And, whether dopamine-release in PL modulates the release of glutamate?

A substantial fraction of VTA glutamergic neurons has been found to release multiple neurotransmitters, including dopamine⁴. Under physiological conditions, the firing of VTA

neurons in response to reward releases glutamate in the PFC, generating fast excitatory postsynaptic potentials (EPSPs) for a rapid response to reward-related stimuli⁶. VTA neurons are thought to use this mechanism to convey temporally precise information to the PFC. In contrast, dopamine-mediated signaling has been shown to gate intrinsic inhibition in PL neurons and modulate the excitability of PFC networks for prolonged periods^{7,8}. Studies on the mesolimbic system suggest that the disruption of the co-release of dopamine and glutamate from VTA neurons reduces mice's responses to psychostimulants^{9,10}. However, the extent to which the mesocortical glutamatergic pathway independently modulates pain or interacts with dopamine signaling remains unclear. To address this, we employed a newly developed viral-based CRISPR/Cas9 approach to selectively deplete TH, thereby preventing the production and release of dopamine from VTA glutamatergic neurons projecting to the PL¹¹. Our results demonstrate that the photoactivation of VTA glutamatergic terminals in the PL effectively alleviates aversion and allodynia, even in the absence of dopamine signaling (**Fig. 4**). These findings highlight the existence of a distinct mesocortical pathway that can modulate neuropathic pain without relying on simultaneous dopamine release. This is consistent with a previous report indicating that the VTA glutamatergic pathway can drive positive reinforcement independent of dopamine signaling¹².

One limitation of the current study is that we did not measure the level of glutamate release in the PL of mice with or without dopamine depletion in VTA-PL projections. It remains unclear whether dopamine release in the PL can modulate the release of glutamate. Within the PL, we found that neurons receiving projections from the VTA are predominantly located in the deep layers (**new Supplementary Fig. 5**). Within the deep-layer PL, there is an enrichment of D1 receptor expression in pyramidal neurons, some of which project to the VTA¹³. These VTA-projecting pyramidal neurons have been shown to modulate VTA neurons that project back to the PFC¹⁴. Thus, it is possible that dopamine release in the PL could modulate the release of glutamate. Future studies of glutamate dynamics in the living brain using the glutamate sensor iGluSnFR may help address this question. It will also help elucidate whether VTA glutamatergic and dopaminergic pathways function as parallel or overlapping systems in modulating pain sensation and affect.

In the revised manuscript, we have included the above-discussed points. We thank the reviewer for these constructive comments.

5. The changes of overall VTA glutamatergic activity in mice with neuropathic pain should be measured?

As suggested, we have conducted new experiments to measure the overall activity of VTA glutamatergic neurons in mice with neuropathic pain. In this experiment, we injected AAV expressing Cre-dependent GCaMP6s into the VTA of *Vglut2*^{ires-Cre} mice and implanted a

gradient-index (GRIN) lens above the VTA to enable *in vivo* Ca^{2+} imaging (**new Supplementary Fig. 2a**). Compared to the pre-SNI baseline, we observed a slight decrease in the somatic activity of VTA glutamatergic neurons two weeks after SNI ($7.57 \pm 0.42\%$ vs. $6.58 \pm 0.42\%$, $P = 0.0103$; **new Supplementary Fig. 2b, c**). No significant changes in neuronal activity were observed in the sham group before and after surgery (**new Supplementary Fig. 2c**). Notably, the distribution of surgery-induced changes in neuronal activity appears to be broader in the SNI group as compared to the sham group (**new Supplementary Fig. 2d, e**), with some cells increasing activity and others decreasing activity.

Previous studies have indicated that NAc-projecting VTA glutamatergic neurons increase activity in chronic pain conditions². Additionally, our axonal imaging results also revealed projection-specific effects of neuropathic pain on VTA glutamatergic terminals, with a decrease of Ca^{2+} activity observed in the PL (**Fig. 1g, h**), but no changes identified in the ACC (**new Supplementary Fig. 9**). Therefore, although the overall activity of VTA glutamatergic neurons is decreased in mice with neuropathic pain, these neurons may exhibit distinct responses to neuropathic pain based on their specific circuits.

New Supplementary Fig. 2: Analysis of the somatic activity of VTA glutamatergic neurons in mice with neuropathic pain.

a, Experimental design for expressing GCaMP6 in VTA glutamatergic neurons and *in vivo* two-photon (2P) Ca^{2+} imaging in the VTA through a GRIN lens. **b**, Representative two-photon images of VTA glutamatergic neurons expressing GCaMP6s (left) and their corresponding Ca^{2+}

traces before and two weeks after SNI (right). Scale bar, 20 μm . **c**, Somatic Ca^{2+} activity before and two weeks after sham or SNI surgery (sham, $P = 0.1636$, $n = 112$ cells from ten mice; SNI, $P = 0.0103$, $n = 119$ cells from nine mice). **d**, Changes in somatic Ca^{2+} activity after sham or SNI surgery ($P = 0.0998$). **e**, Distribution plot of data shown in **d**. The distribution of SNI-induced changes in neuronal activity appears to be broader compared to the sham group, suggesting that VTA glutamatergic neurons may exhibit distinct responses to neuropathic pain based on their specific circuits. Summary data are presented as mean \pm S.E.M. * $P < 0.05$; ns, not significant; by Wilcoxon test (**c**) or Mann-Whitney test (**d**).

6. Whether has functional monosynaptic connection between the VTA glutamatergic neurons and PL glutamatergic neurons should be tested with electrophysiological method?

We thank the reviewer for the constructive comment. Indeed, previous electrophysiological studies have consistently demonstrated that glutamate release from the VTA depolarizes PFC neurons, eliciting monosynaptic EPSPs that are sensitive to glutamate receptor antagonists^{7,15}. Built upon this, our study provides additional evidence supporting monosynaptic connections in VTA-PL glutamatergic circuits. By using viral infection to label pre-synapses, we confirmed the presence of VTA glutamatergic presynaptic terminals in the PL (**Fig. 1d, e**), which were responsive to sensory stimulation (**new Supplementary Fig. 1c, d**). Through a virus-mediated transsynaptic tagging approach¹⁶, we showed that the majority of (~75%) PL neurons receiving VTA glutamatergic presynaptic inputs are excitatory (**Fig. 5**) and are primarily located in the deep layers of PL (**new Supplementary Fig. 5**). Furthermore, optogenetic activation of VTA-PL glutamatergic terminals increased the axonal outputs of postsynaptic PL neurons projecting to the ACC (**Fig. 7c, d**). These findings, along with prior electrophysiological studies^{4,5,17}, collectively establish the presence of functional monosynaptic connectivity in the VTA-PL glutamatergic circuits.

7. Which receptor (NMDAR or AMPAR) in PL glutamatergic neurons mediates this effect of pain regulation?

NMDA and AMPA receptors play important roles in the development of chronic pain¹⁸. In previous studies using *in vivo* electrophysiological and electrochemical methods, it has been demonstrated that the mesocortical system generates a fast, non-dopamine-mediated postsynaptic response in the PFC. This glutamate-initiated response can be blocked by the AMPA receptor antagonist CNQX^{7,15}. Additionally, functional magnetic resonance imaging studies reveal that electrical stimulation of VTA can induce blood-oxygen-level dependent responses in the PFC, which can be abolished by the NMDA receptor antagonist MK801^{19,20,21}. Within the PL, changes in NMDA and AMPA receptor levels have been extensively documented under chronic pain conditions^{22,23,24,25}. Taken together, these findings suggest that both NMDA and AMPA

receptors may contribute to the pain-modulatory effects of PL glutamatergic neurons. We thank the reviewer for the thoughtful comment.

8. Whether direct VTA glutamatergic neurons inverters into the ACC and its functional role in the development of chronic neuropathic pain?

The reviewer raised an intriguing hypothesis--whether VTA glutamatergic neurons could directly affect ACC function, aside from using PL neurons as a relay. To explore this possibility, we have conducted new experiments to examine the Ca^{2+} activity of VTA-ACC glutamatergic axons in mice with neuropathic pain. Specifically, we injected an AAV encoding axon-targeted GCaMP6s under Cre-dependent control into the VTA of $Vglut2^{IRES-Cre}$ mice (**Supplementary Fig. 9a**). Two weeks after SNI or sham surgery, we performed *in vivo* Ca^{2+} imaging in the ACC of these mice. In contrast to the observations in VTA-PL glutamatergic terminals (**Fig. 1g, h**), we found no significant difference in Ca^{2+} activity of VTA-ACC glutamatergic terminals in resting SNI mice compared to sham mice ($33.67 \pm 0.87\%$ vs. $35.66 \pm 0.99\%$, $P = 0.3362$, **new Supplementary Fig. 9b-d**). These results indicate that there are no alterations in the VTA-ACC glutamatergic pathway in mice with neuropathic pain.

Taken together, our data suggest that VTA glutamatergic neurons do not operate as a single functional unit, and their function is likely determined by the circuit of which they are a part and the behaviors that depend on the operation of that circuit. We thank the reviewer for suggesting this experiment.

New Supplementary Fig. 9: VTA-ACC glutamatergic activity is not altered in mice with neuropathic pain.

a, Experimental design for expressing GCaMP6 in the axons of VTA glutamatergic neurons and performing *in vivo* Ca²⁺ imaging in the ACC two weeks after sham or SNI surgery. **b**, Representative two-photon images and fluorescence traces of ACC-projecting VTA glutamatergic axons expressing GCaMP6. Scale bar, 10 μ m. **c**, Ca²⁺ activity in VTA-ACC glutamatergic terminals ($P = 0.3362$; $n = 314/5, 319/6$ axon segments/mice). **d**, Distribution plot of data shown in **c**. Summary data are presented as mean \pm S.E.M. ns, not significant; by Mann-Whitney test.

Reviewer #2 (Remarks to the Author):

Li & Yang report in this manuscript that spared nerve injury (SNI) leads to a reduction in the activity of VTA-VGluT2⁺ terminals in the prelimbic cortex. They report using optogenetic and chemogenetic strategies that activation of this projection alleviates pain systems and increases the activity of glutamatergic PL neurons synaptically linked to VTA-VGluT2⁺ neurons, whereas inhibition of this circuit induces neuropathic pain-like behaviors in intact mice and reduced the activity of PL neurons. Using CRISPR/Cas9 deletion of tyrosine hydroxylase, they demonstrate that dopamine co-release is not involved in the alleviation of pain-like behaviors via activation of this circuit. Finally, they find that activation of VTA-VGluT2-PL leads to increased activity of PL terminals in the ACC and that optogenetic activation of PL/VTA-PL terminals in the ACC similarly alleviates pain-like behaviors. Overall, this is a well-conducted series of experiments elucidating the role of glutamatergic signaling from the anterior VTA, a region where $\sim 2/3$ of neurons are VGluT2⁺, to the PL cortex in neuropathic pain.

We thank the reviewer for encouraging and constructive comments on our work.

There are some concerns with interpretation of certain experimental results and methods that need to be addressed before the manuscript is published.

1) The authors state that they used the rtPP and CPP assays to assess pain aversion (line 109) and interpret the induced preference in both assays to indicate pain relief in mice with SNI. This is puzzling given that the canonical interpretation of this type of results is that the manipulation under investigation is rewarding/reinforcing. How these results relate to pain relief is therefore unclear, especially since the authors do not report or cite previous literature showing the effects of these manipulations in intact mice.

We thank the reviewer for raising this concern. As the reviewer rightly pointed out, rtPP and CPP are best known for their application in investigating rewarding and reinforcing behaviors. Additionally, these assays have been adapted in preclinical pain research to assess ongoing pain

or related aversion in rodents^{26, 27, 28}. The underlying rationale is that the mice with spontaneous or ongoing pain may exhibit a preference for the chamber paired with interventions that provide pain relief (e.g., analgesic drugs or, in this case, optical stimulation to activate a specific brain circuit). We have now referenced relevant literature using these tests for ongoing pain assessment.

As the reviewer pointed out correctly, the original submission did not rule out the possibility that the activation of VTA-PL glutamatergic terminals could induce a rewarding/reinforcing effect in naïve mice without neuropathic pain. To address this concern, we have conducted additional experiments to assess the effects of optical manipulation in sham control mice (*i.e.*, Sham+hChR2) (**new Fig. 2a-e**). Our results show that the activation of the VTA-PL pathway in control mice without neuropathic pain does not induce any preference effects in both the rtPP and CPP tests. Together with our data from the SNI+hChR2 and SNI+mCherry groups, these results indicate that the activation of the VTA-PL pathway can alleviate ongoing pain in a mouse model of neuropathic pain. We thank the reviewer for suggesting this important control experiment.

Fig. 2: Optogenetic activation of VTA glutamatergic terminals in the PL alleviates pain-associated behaviors.

a, Experimental design to express hChR2 or mCherry (control) in VTA glutamatergic neurons and stimulate their terminals in the PL. **b**, Schematic of real-time place preference (RT-PP) test for assessing ongoing pain. **c**, Preference to stay in the light stimulation-paired chamber for SNI mice expressing mCherry or hChR2 and sham mice expressing hChR2 ($P = 0.0036$ and 0.0137 for SNI+hChR2 vs. SNI+mCherry and sham+hChR2, respectively). **d**, Schematic of conditioned place preference (CPP) test for assessing ongoing pain. **e**, Preference to stay in the light stimulation-paired chamber for SNI mice expressing mCherry or hChR2 and sham mice expressing hChR2 ($P = 0.0077$ and 0.0061 for SNI+hChR2 vs. SNI+mCherry and sham+hChR2, respectively). In **c** and **e**, $n = 6, 8, 5$ mice. **f**, Schematic of mechanical and thermal tests with/without light stimulation in SNI mice. **g–i**, Measurements of nociceptive thresholds in SNI mice expressing mCherry or hChR2. Activation of VTA-PL glutamatergic terminals increases the animals' mechanical ($P = 0.0078$), hot ($P = 0.0078$), and cold ($P = 0.0078$) thresholds in the limb ipsilateral to SNI. In **g–i**, $n = 6, 8$ mice for mCherry, hChR2 respectively. Summary data are presented as mean \pm S.E.M. * $P < 0.05$, ** $P < 0.01$; ns, not significant; by Dunn's multiple comparisons tests (**c**, **e**) or Wilcoxon test (**g–i**).

2) The authors report that they conducted their optogenetic studies using 1mW (Blue) or 1-2mW (yellow) light stimulation. These are extremely low stimulation intensities for terminal modulation and inconsistent with the generally reported ~ 10 mW stimulation for optogenetic terminal stimulation. This concern would be alleviated by the author demonstrating that activation resulted in cFos expression in the PFC of the experimental cohorts.

We thank the reviewer for the thoughtful comment. As suggested, we have conducted c-Fos staining to verify the efficacy of optogenetic activation of the VTA-PL glutamatergic pathway. Specifically, two weeks after the viral injection, we applied blue light to the mouse PL. One hour after light stimulation, the brain section containing PL was collected and immunostained for c-Fos, a marker of neuronal activity. Subsequent confocal imaging revealed a significant increase in c-Fos⁺ cells in the PL of mice expressing hChR2 compared to those expressing mCherry (**new Supplementary Fig. 3**). These data validate the effectiveness of our protocols for optogenetic terminal stimulation.

New Supplementary Fig. 3: Optogenetic stimulation of VTA-PL glutamatergic terminals activates PL.

a, Experimental design for expressing hChR2 or mCherry (control) in VTA glutamatergic (vGluT2⁺) neurons and stimulating their terminals in the PL. **b**, Representative images of PL immunostained for c-Fos, a neuronal activity marker. **c**, Optical stimulation of VTA-PL glutamatergic terminals expressing hChR2 results in increased c-Fos expression in the PL ($P < 0.0001$, $n = 10$, 9 slices from three mice per group). Summary data are presented as mean \pm S.E.M. **** $P < 0.0001$; by Mann-Whitney test.

Minor Comments:

1) In which cortical layer(s) were PL_{VTA-PL} neurons are located?

We have now quantified the distribution of PL neurons that receive VTA inputs (**new Supplementary Fig. 5**). Most (~85%) of these PL_{VTA-PL} neurons are located in the deep layers (e.g., >300 μ m subpial).

New Supplementary Fig. 5: Distribution of PL neurons receiving VTA inputs.

a, Experimental design for expressing mCherry in PL neurons receiving VTA projections (PL_{VTA-PL} neurons). **b**, Representative confocal image showing mCherry-labelled PL_{VTA-PL} neurons. **c**, Distribution of mCherry⁺ somas across layers of PL (*n* = 9 slices from four mice). Scale, 100 μm.

2) It is not completely clear what the authors mean by neuron deactivation. Does this imply that the neurons are simply less active generally? Clearly the activity of these neurons can be increased as shown in the results of Figure 7. Clarification of this term would be useful to prevent ambiguity in understanding the interpretation of these results.

The reviewer is correct; even though VTA glutamatergic neurons generally exhibit less activity in neuropathic pain conditions, they can still respond to painful stimulation (**new Supplementary Fig. 1c, d**). As advised, we have removed the term “deactivation” in the revised manuscript.

We thank the reviewer again for all the time and effort in helping us improve our manuscript.

References

1. Root DH, Mejias-Aponte CA, Qi J, Morales M. Role of glutamatergic projections from ventral tegmental area to lateral habenula in aversive conditioning. *J Neurosci* **34**, 13906-13910 (2014).
2. Abdul M, *et al.* VTA-NAc glutaminergic projection involves in the regulation of pain and pain-related anxiety. *Front Mol Neurosci* **15**, 1083671 (2022).
3. Wang HL, Qi J, Zhang S, Wang H, Morales M. Rewarding Effects of Optical Stimulation of Ventral Tegmental Area Glutamatergic Neurons. *J Neurosci* **35**, 15948-15954 (2015).
4. Yamaguchi T, Wang HL, Li X, Ng TH, Morales M. Mesocorticolimbic glutamatergic pathway. *J Neurosci* **31**, 8476-8490 (2011).
5. Hnasko TS, Hjelmstad GO, Fields HL, Edwards RH. Ventral tegmental area glutamate neurons: electrophysiological properties and projections. *J Neurosci* **32**, 15076-15085 (2012).
6. Traynelis SF, *et al.* Glutamate receptor ion channels: structure, regulation, and function. *Pharmacol Rev* **62**, 405-496 (2010).
7. Lavin A, Nogueira L, Lapish CC, Wightman RM, Phillips PE, Seamans JK. Mesocortical dopamine neurons operate in distinct temporal domains using multimodal signaling. *J Neurosci* **25**, 5013-5023 (2005).
8. Buchta WC, Mahler SV, Harlan B, Aston-Jones GS, Riegel AC. Dopamine terminals from the ventral tegmental area gate intrinsic inhibition in the prefrontal cortex. *Physiol Rep* **5**, (2017).
9. Hnasko TS, *et al.* Vesicular glutamate transport promotes dopamine storage and glutamate corelease in vivo. *Neuron* **65**, 643-656 (2010).
10. Birgner C, *et al.* VGLUT2 in dopamine neurons is required for psychostimulant-induced behavioral activation. *Proc Natl Acad Sci U S A* **107**, 389-394 (2010).
11. Hunker AC, Soden ME, Krayushkina D, Heymann G, Awatramani R, Zweifel LS. Conditional Single Vector CRISPR/SaCas9 Viruses for Efficient Mutagenesis in the Adult Mouse Nervous System. *Cell Rep* **30**, 4303-4316 e4306 (2020).
12. Zell V, *et al.* VTA Glutamate Neuron Activity Drives Positive Reinforcement Absent Dopamine Co-release. *Neuron* **107**, 864-873 e864 (2020).

13. Anastasiades PG, Boada C, Carter AG. Cell-Type-Specific D1 Dopamine Receptor Modulation of Projection Neurons and Interneurons in the Prefrontal Cortex. *Cereb Cortex* **29**, 3224-3242 (2019).
14. Fields HL, Hjelmstad GO, Margolis EB, Nicola SM. Ventral tegmental area neurons in learned appetitive behavior and positive reinforcement. *Annu Rev Neurosci* **30**, 289-316 (2007).
15. Perez-Lopez JL, Contreras-Lopez R, Ramirez-Jarquín JO, Tecuapetla F. Direct Glutamatergic Signaling From Midbrain Dopaminergic Neurons Onto Pyramidal Prefrontal Cortex Neurons. *Front Neural Circuits* **12**, 70 (2018).
16. Zingg B, *et al.* AAV-Mediated Anterograde Transsynaptic Tagging: Mapping Corticocollicular Input-Defined Neural Pathways for Defense Behaviors. *Neuron* **93**, 33-47 (2017).
17. Hnasko TS, Edwards RH. Neurotransmitter corelease: mechanism and physiological role. *Annu Rev Physiol* **74**, 225-243 (2012).
18. Zhuo M. Ionotropic glutamate receptors contribute to pain transmission and chronic pain. *Neuropharmacology* **112**, 228-234 (2017).
19. Zheng N, *et al.* Investigations of brain-wide functional and structural networks of dopaminergic and CamKIIalpha-positive neurons in VTA with DREADD-fMRI and neurotropic virus tracing technologies. *J Transl Med* **21**, 543 (2023).
20. Brocka M, *et al.* Contributions of dopaminergic and non-dopaminergic neurons to VTA-stimulation induced neurovascular responses in brain reward circuits. *Neuroimage* **177**, 88-97 (2018).
21. Helbing C, Brocka M, Scherf T, Lippert MT, Angenstein F. The role of the mesolimbic dopamine system in the formation of blood-oxygen-level dependent responses in the medial prefrontal/anterior cingulate cortex during high-frequency stimulation of the rat perforant pathway. *J Cereb Blood Flow Metab* **36**, 2177-2193 (2016).
22. Kelly CJ, Huang M, Meltzer H, Martina M. Reduced Glutamatergic Currents and Dendritic Branching of Layer 5 Pyramidal Cells Contribute to Medial Prefrontal Cortex Deactivation in a Rat Model of Neuropathic Pain. *Front Cell Neurosci* **10**, 133 (2016).

23. Guida F, *et al.* Palmitoylethanolamide reduces pain-related behaviors and restores glutamatergic synapses homeostasis in the medial prefrontal cortex of neuropathic mice. *Mol Brain* **8**, 47 (2015).
24. Medeiros P, *et al.* N-methyl-D-aspartate Receptors in the Prelimbic Cortex are Critical for the Maintenance of Neuropathic Pain. *Neurochem Res* **44**, 2068-2080 (2019).
25. Li A, *et al.* Disrupted population coding in the prefrontal cortex underlies pain aversion. *Cell Rep* **37**, 109978 (2021).
26. Chiang MC, Nguyen EK, Canto-Bustos M, Papale AE, Oswald AM, Ross SE. Divergent Neural Pathways Emanating from the Lateral Parabrachial Nucleus Mediate Distinct Components of the Pain Response. *Neuron* **106**, 927-939 e925 (2020).
27. Zhou K, *et al.* Reward and aversion processing by input-defined parallel nucleus accumbens circuits in mice. *Nat Commun* **13**, 6244 (2022).
28. Sufka KJ. Conditioned place preference paradigm: a novel approach for analgesic drug assessment against chronic pain. *Pain* **58**, 355-366 (1994).

REVIEWERS' COMMENTS

Reviewer #1 (Remarks to the Author):

The authors have already answered my concerns by some new data. The manuscript was well revised. I have no further concerns.

Reviewer #2 (Remarks to the Author):

In this revised manuscript, the authors have performed a significant number of new experiments that clarify and strengthen the data and conclusion from the original submission. These experiments have addressed all of this reviewer's previous concerns.

Responses to the reviewers' comments

Reviewer #1 (Remarks to the Author):

The authors have already answered my concerns by some new data. The manuscript was well revised. I have no further concerns.

Reviewer #2 (Remarks to the Author):

In this revised manuscript, the authors have performed a significant number of new experiments that clarify and strength then the data and conclusion from the original submission. These experiments have addressed all of this reviewer's previous concerns.

We sincerely thank all reviewers for providing critiques of our work and helping us strengthen our findings.